# The influence of freshwater inflow and seascape context on occurrence of juvenile spotted seatrout *Cynoscion nebulosus* across a temperate estuary

**Shannon D. Whaley**[1]*, **Colin P. Shea**[1], **E. Christine Santi**[1], **David A. Gandy**[2]

**1** Florida Fish and Wildlife Conservation Commission, Fish and Wildlife Research Institute, St Petersburg, Florida, United States of America, **2** Florida Fish and Wildlife Conservation Commission, Fish and Wildlife Research Institute, Apalachicola Bay Field Laboratory, Eastpoint, Florida, United States of America

* Shannon.Whaley@myFWC.com

**Data Availability Statement:** All data used in this analysis can be accessed here: https://f50006a.eos-intl.net/F50006A/OPAC/Details/Record.aspx?BibCode=5847989.

## Abstract

Spotted seatrout, a popular recreational sport fish in the southeastern United States, are affected by freshwater flow conditions and the availability of estuarine habitat. However, the relative influence of these factors, particularly on early life stages of seatrout, remains uncertain. We used generalized linear models to quantify relationships between the probability of encountering juvenile spotted seatrout during seine surveys and various factors, including freshwater inflow conditions, the availability and richness of estuarine habitats (seagrass, salt marsh, oyster beds) around (400-m radius) fish collection sites (seascape-scale context), as well as distance to the nearest inlet to the Gulf of Mexico (estuary-scale context) across shallow waters (< 1.5 m depth) of Apalachicola Bay, Florida. Modelling results showed a consistent positive correlation between seagrass area and the probability of encountering juvenile seatrout (all four size classes from 15mm–200mm Standard Length (SL)). The probability of encountering the two smallest juvenile seatrout size classes (15–50mm and 51–100mm SL) was also related to freshwater inflow conditions, particularly within a 3-month period prior to and including peak recruitment. Freshwater inflow may affect early life stages by influencing passive transport of eggs and larvae, planktonic food availability, and predation pressure through increases in turbidity. In contrast, encounter probabilities of the two larger size classes (101–150mm and 151–200mm) were unrelated to freshwater inflow. Inflow-related processes may be less important to the larger juveniles as they have typically settled out of the plankton into benthic habitats which provide refuge from predation and abundant benthic food sources which are not as closely tied to freshwater inflow effects. In addition, models revealed that occurrence of the larger juveniles was related to the availability of nearby habitat types such as oyster beds and salt marshes, suggesting that increased mobility as seatrout grow may allow them to use nearby habitat types as additional sources of food and refuge. These results add to a growing body of literature aimed at understanding the influence of freshwater inflow as well as seascape context on vulnerable juvenile life stages of fishery species to provide more informed strategies for freshwater inflow management and habitat conservation.

**Funding:** This research was financially supported by the US Fish and Wildlife Service, Sport Fish Restoration Grant F-66 (SDW) and US Fish and Wildlife Service, Sport Fish Restoration Grant F-43 (DAG). The funders had no role in study design, data collection and analysis, decision to publish, or preparation of the manuscript.

**Competing interests:** The authors have declared that no competing interests exist.

## Introduction

Riverine flow into estuarine and coastal ecosystems is linked to at least one life stage for many fish species that support important recreational and commercial fisheries worldwide [1]. Fluctuations in riverine flows can influence productivity of estuarine-dependent fishes through several mechanisms, including changes to circulation patterns, productivity through delivery of land-based sources of nutrients, and spatial distribution and inundation of wetland floodplains (e.g., salt marshes) and subtidal habitats such as seagrass beds, and oyster beds [2, 3].

The availability of estuarine habitat types has been positively related to productivity of early life history stages for many fish species [4, 5]. Habitat-use patterns have largely been examined by relating species occurrence or relative abundance with the presence or absence of individual habitat types within the sample location [6]. However, many juveniles are highly mobile, and may use multiple nearby subtidal and intertidal habitats [7–11]. Availability of multiple biogenic habitats (habitat richness and area) throughout the nearby seascape, or seascape context, can be more important than the presence of a particular habitat type in determining habitat function for juvenile fishes [12–14]. The term "seascape nurseries" has been used to represent habitat mosaics containing multiple habitat types in close proximity that are functionally connected because they are frequently used by highly mobile juveniles of fishery species [6]. Quantifying habitat patches (area or edge metrics) and seascape context relative to both the presence of multiple habitat types nearby (habitat richness) as well as location relative to the connection to the marine environment (nearest inlet) may help identify and manage seascape nurseries important to juveniles of fishery species [15–17].

Spotted seatrout *Cynoscion nebulosus* is a popular recreational species in the southeastern United States [18]. Spotted seatrout complete their life cycle within their natal estuary; therefore, populations within individual estuaries are somewhat genetically distinct [19], and habitat-use patterns, particularly of juveniles, can also differ among estuaries [20–28]. Generally, juvenile spotted seatrout are highly mobile and may use multiple habitat types nearby [29, 30]. Juvenile seatrout are strongly associated with seagrass habitat [15, 22, 24–27, 31], yet they can also be found in relatively high densities in a variety of habitat types, including salt marsh edges [21] and backwater areas [20]. Therefore, habitat use patterns of juveniles may be related to seascape context, either the area of individual habitat types (particularly seagrass) or habitat richness. Few studies have examined seascape context relative to juvenile seatrout, although higher catch rates of juvenile seatrout were found in areas containing multiple nearby habitat types compared with areas with only one habitat type in a North Carolina estuary [9]. In addition, the area of seagrass habitat, measured at broad scales (1600-m radius), was related to juvenile seatrout occurrence in an estuary in southwestern Florida [15]. Given the importance of habitat on juvenile seatrout, habitat area and the presence of nearby habitat types may also explain habitat use patterns in other estuaries.

In addition to habitat, the distribution or relative abundance of juvenile spotted seatrout has been related to freshwater inflow conditions in several estuaries within the southeastern United States [28, 30–32]. However, the influence of freshwater inflow (inflow) on spatial distribution of juvenile spotted seatrout appears to vary among life stages [28] and depending upon the temporal scales at which inflow is measured [32, 33]. In Tampa Bay, for example, inflow conditions measured over a 12-month period prior to fish sampling were positively related to distribution and abundance of early juveniles (15–50mm SL), yet the influence of inflow diminished as the juveniles grew larger (51–100mm SL, [28]). In the Suwannee River estuary, a positive relationship between relative abundance of seatrout juveniles (less than 100mm SL) and inflow was found when inflow was measured over a three-month time lag [32] but this positive relationship was weaker when inflow was measured on an annual basis

[33]. Identifying the most relevant temporal scale to measure freshwater inflow conditions as well as understanding which life stages are influenced by inflow conditions may provide important clues to underlying mechanisms responsible for these patterns, as well as help develop water management strategies to benefit this important fishery species.

To assess the roles of freshwater inflow and seascape context in influencing juvenile seatrout distribution patterns, we examined the occurrence of juvenile spotted seatrout *Cynoscion nebulosus* in shallow waters of Apalachicola Bay, Florida (USA) during a period of relatively high seasonal occurrence. Peak season of juvenile occurrence was defined as June-November for small juveniles (15–50mm and 51–100mm SL) and August-December for larger juveniles (101–150mm and 151–200mm SL). With respect to seatrout encounter probabilities for each of four size classes (15–50mm, 51–100mm, 101–150mm, 151–200mm SL), the objectives of the study were to use mixed effects logistic regression to: (1) determine the relative importance of freshwater inflow conditions at several temporal scales (1-month, 3-months, 6-months, 12-months), (2) determine the influence of habitat area (seagrass, salt marsh, and oyster beds) and number of habitat types (habitat richness) within a 400m radius of fish collection sites (seascape-scale context), distance to the nearest inlet (estuary-scale context), and water depth, (3) determine the extent to which freshwater inflow conditions affect seatrout encounter probabilities at seascape and estuary-wide spatial scales, and (4) use the best approximating models to map spatial patterns of juvenile seatrout occurrence.

## Methods

### Ethics statement

Specimen collections were conducted via standard protocols by the Florida Fish and Wildlife Research Institute (FWRI) Fisheries Independent Monitoring (FIM) program. These protocols were authorized by the Florida Fish and Wildlife Conservation Commission for state fisheries and conservation research. Every effort was made to reduce stress and not harm captured fish, before releasing. No protected species were sampled.

### Study area

Apalachicola Bay (Fig 1), located on the northeastern Gulf Coast, is one of the least populated coastal areas in Florida which likely contributes to its relatively high overall water quality [34]. The Apalachicola River drains the Apalachicola-Chattahoochee-Flint River Basin, a large 50,500 km$^2$ watershed, and contributes the majority of freshwater to the Bay. Northeastern Florida's climate is considered subtropical with mild winters and year-round rainfall (no specific dry season). Apalachicola Bay is connected to the Gulf of Mexico through three natural tidal inlets (Indian Pass, West Pass, East Pass) and one man-made inlet (Government Cut a.k.a. Sike's Cut, Fig 1). Natural shorelines in the estuary proper are dominated by salt marshes (primarily black needlerush *Juncus roemerianus*), and oyster reefs *Crassostrea virginica* and seagrass beds (principally shoal grass *Halodule wrightii*, turtle grass *Thalassia testudinum*, and manatee grass *Syringodium filiforme*) are common in shallow-water areas throughout the Bay (Fig 2).

### Fish collection

Juvenile spotted seatrout were collected monthly during daylight hours from 2001–2018 by the State of Florida's Fisheries-Independent Monitoring Program (FIM; Florida Fish and Wildlife Conservation Commission, Fish and Wildlife Research Institute, Fig 1). Monthly samples (64 samples/month on average) were selected via a stratified random sampling design. In this

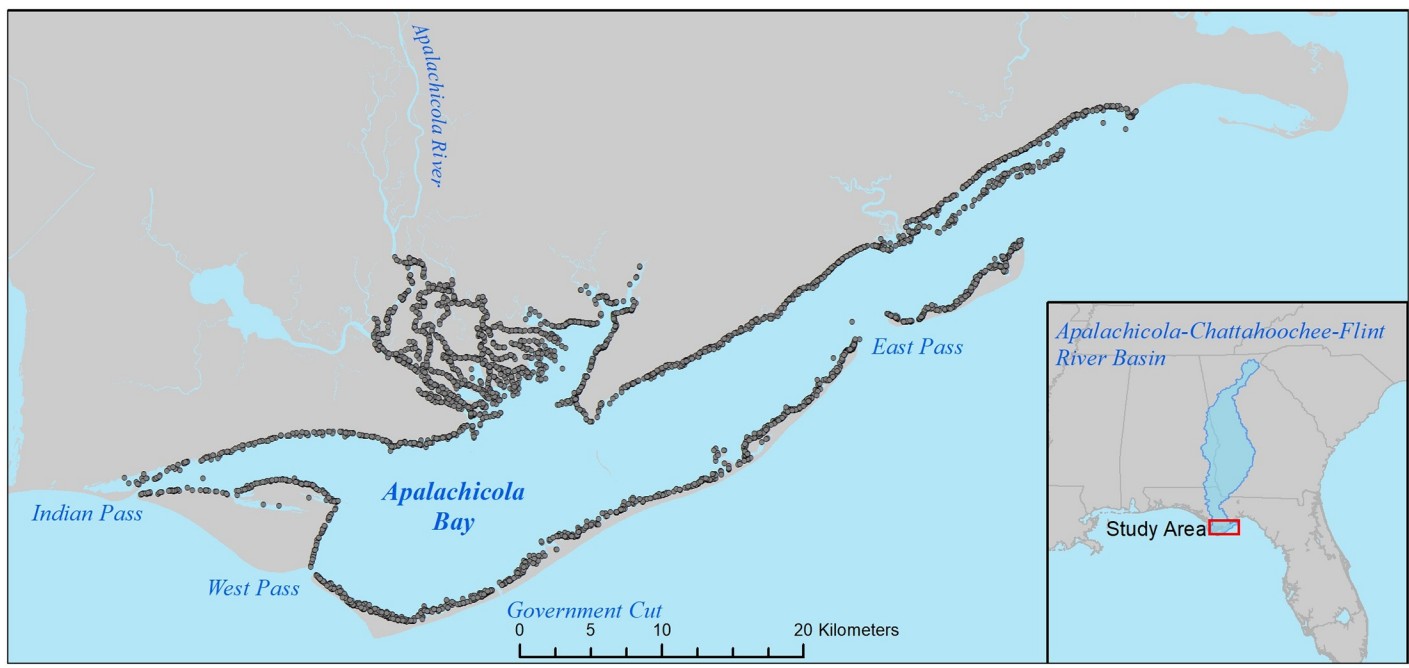

**Fig 1.**

design, the Apalachicola study area was divided into several large spatial zones. Each zone was divided into 1 min latitude × 1 min longitude grid cells, and these were randomly selected each month for sampling. Smaller juveniles (15–100mm Standard Length) were collected with a 21.3-m x 1.8-m center-bag seine (3.2-mm stretched mesh) and deployed in water depths

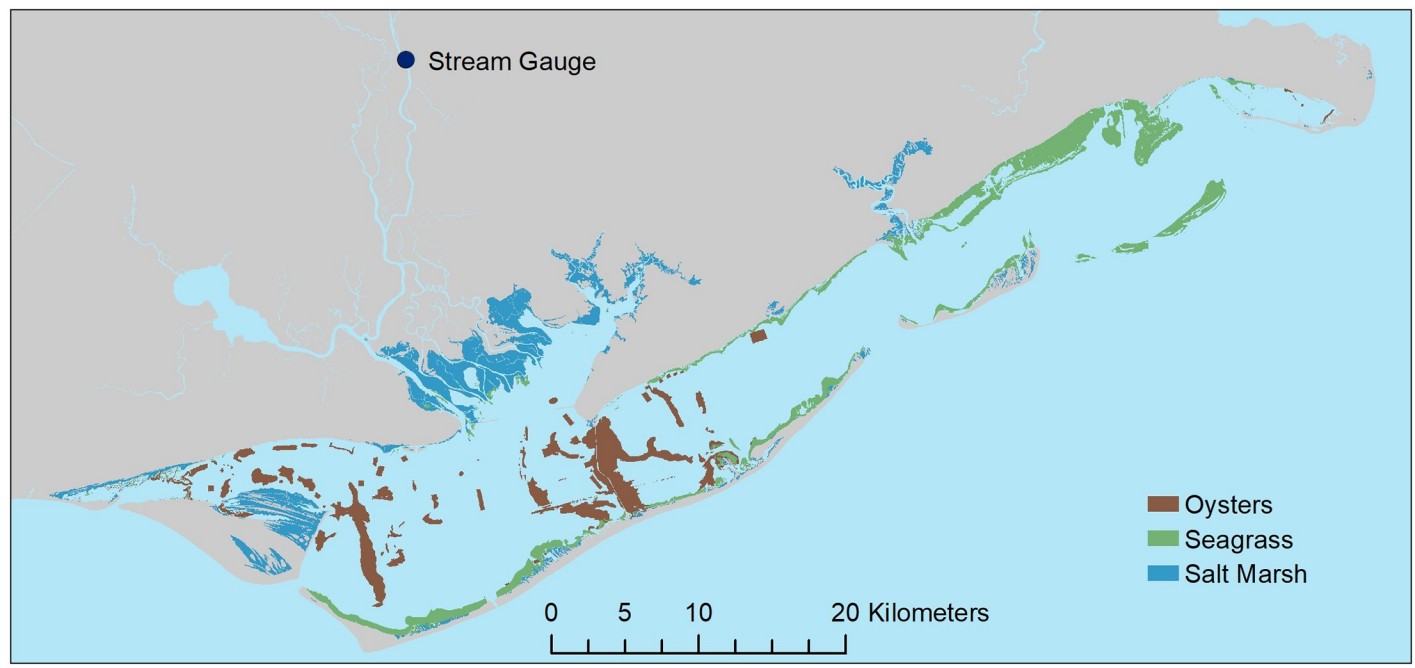

**Fig 2.**

of $\leq$ 1.8-m (rivers) or $\leq$ 1.5-m (estuary) from June 2003 through November 2018. Larger juveniles (100–200mm SL) were collected with a 183-m × 2.5-m center bag haul seine (38-mm stretched mesh) in water depths of $\leq$ 2.5 m from August 2001 through December 2018. Sampling methods are described in detail in [32, 35, 36]. Time periods representing relatively high seasonal occurrence (peak recruitment) for each size class were determined from monthly length-frequency histograms. Peak spawning season in Apalachicola Bay occurs from April to August [19]. Peak recruitment season for the two smaller juvenile size classes (15–50mm SL and 51–100mm SL) occurred from June to November (3,094 unique samples). Peak recruitment season for the larger two juvenile size classes (101–150mm SL, 151–200mm SL) occurred from August to December (1,782 unique samples). Data for each sample within these peak periods were converted into presence or absence to examine the predominant factors influencing the probability of encountering seatrout size classes.

## Seascape and estuary-scale habitat features

We compiled the most comprehensive regional Geographic Information System (GIS) data layers for salt marshes, seagrasses, and oyster beds. We extracted salt marsh habitat data from digital land use and land cover data (1:12,000-scale) acquired from the Northwest Florida Water Management District [37]. For seagrass beds, we used data interpreted using natural color aerial photography collected in October 2010 [38]. These spatial coverages of seagrass [38] and salt marsh [39] habitat represent the best available data sources and were relatively stable over the fish sampling period. Oyster bed data were obtained from a statewide oyster compilation data set for Apalachicola Bay [40], for which multiple datasets of varying resolutions and dates ranging from 1970 to 2018 were combined to achieve full coverage of the estuary. Although coverage of living oyster beds in Apalachicola Bay has declined dramatically in recent years, oyster shells have remained and likely continue to provide habitat for small fishes by providing protection from predation. Therefore, oyster maps used in this study are based on coverage of oyster habitat (both living and dead). Depth data in meters, referenced to the North American Vertical Datum of 1988, were obtained from NOAA's Continuously Updated Digital Elevation Model (CUDEM) at 1/9 Arc-Second Resolution [41]. We performed quality control checks on all GIS layers including map projection definition, verifying attributes were within described ranges, and topological checks to verify there were no overlapping features.

Using the above-described habitat GIS layers, we calculated the area (m$^2$) of several seascape-scale habitats: salt marsh, oyster beds, discontinuous seagrass beds, continuous seagrass beds, and all seagrass beds (area of discontinuous and continuous habitats combined), within circular zones (radius = 400m) surrounding each 10m ×10m grid cell within the entire study area. Additionally, we calculated mean depth and habitat richness (i.e., the sum of three binary variables, salt marsh presence, seagrass presence, oyster bed presence) within the 400m radius zone. To examine potential relationships between connectivity to the marine environment (i.e., estuary-scale features) and seatrout occurrence, we also calculated the shortest distance (meters) from each grid cell in the study area to the nearest major inlet (Fig 1) connecting the bay to the Gulf of Mexico using the cost-distance function provided in the ArcGIS software package [42], using the shoreline as a barrier so that distances are constrained to be over the water. In a similar way, we calculated the shortest distance from each grid cell to the nearest river mouth. These calculations resulted in a series of continuous grid surfaces representing each metric across the estuary. Metric values in these grids were associated with each FIM sample location for model development, and subsequently used for spatial modeling of encounter probability across the study area.

## Freshwater inflow

To examine the influence of freshwater inflow patterns on seatrout encounter probabilities, we used a USGS classification where monthly stream flow is compared with a 30-year reference period (1961–1990). Stream flow for the reference period was ranked by magnitude and then classified as above normal ($\geq$ upper quartile), normal (between lower and upper quartiles), or below normal ($\leq$ lower quartile, [43]). We used monthly streamflow values from a gauge located on the mainstem of Apalachicola River (Fig 2) [United States Geological Survey (USGS) gauge Site Number 02359170, Apalachicola River near Sumatra, FL]. We characterized freshwater inflow conditions over several time lags (1-month, 3-months, 6-months, 12-months), prior to and including the month of fish sampling, by calculating the proportion of months within each time lag that were classified as below normal streamflow (% low flow or dry conditions), normal (% normal flow) and above normal streamflow (% high flow or wet conditions). For each time lag, normal flow conditions served as the statistical baseline (i.e., % normal = 1 - % wet + % dry); hence, subsequent logistic regression models (see Statistical analysis, below) only included flow variables associated with wet and dry conditions as predictor variables.

## Statistical analysis

We used mixed effects logistic regression [44] to investigate the influence of estuarine habitat and freshwater inflow on the probability of encountering spotted seatrout during seine sampling. For each seatrout size class, we fit a candidate set of 17 logistic regression models, each representing a different hypothesis regarding the influence of freshwater inflows on seatrout encounter probabilities (Table 1). To minimize the total number of candidate models and focus primarily on the role of freshwater inflows, we included all seascape- and estuary-scale predictors in all models. Twelve of the 17 candidate models represented a different combination of a single freshwater inflow predictor and its interaction with all seascape-scale (four models), all estuary-scale (four models), and all seascape- and estuary-scale (four models) predictor variables; four additional models included a single freshwater inflow predictor and no interactions with seascape- or estuary-scale predictor variables; and the final model was considered a null model that only included seascape and estuary-scale predictors. To avoid multicollinearity, we did not include any two predictor variables with Pearson correlation coefficients greater than |0.5| in the same model. When two predictor variables were collinear, we selected the variable that was most in line with our modelling objectives. To facilitate model-fitting, we standardized the 3-month, 6-month, and 12-month freshwater inflow variables to a mean of zero and standard deviation (SD) of one (the 1-month inflow variable was not standardized as it was simply a binary indicator of wet or dry conditions). To account for potential temporal dependence among seine samples collected in the same month and year, we included a Month × Year random effect associated with the model intercept that was assumed to be normally distributed [44]. Similarly, we accounted for potential spatial autocorrelation among observations by including the latitudinal (for size classes <100mm) or longitudinal (for size classes >100mm) coordinates of each sample location as an additional predictor variable in the model (see below).

Following model fitting, we ranked the relative plausibility of each candidate model using Akaike's Information Criterion (AIC, [45]) with a small-sample bias adjustment ($AIC_c$, [46]), and to facilitate comparisons among models, we calculated relative Akaike weights, which range from zero to one, where the best-approximating model had the highest weight [47]). We note that prior to fitting the 17 candidate models, we used $AIC_c$ to determine if latitude or longitude, which were highly correlated in all data sets, was a better predictor of encounter

**Table 1. Model number, linear predictor of fixed effects included in each model, and model description for the candidate set of mixed effects logistic regression models relating seascape-scale context (habitat richness, and seagrass, saltmarsh, and oyster percent coverage), estuary-scale context (Distance to nearest major inlet), and hydrologic variables to the probability of encountering spotted seatrout during seine and trawl surveys.** All 17 candidate models were fit to data for each of the four size classes: ≤50mm, 51–100mm, 101–150mm, and 151–200mm. All models also included a Month × Year random intercept, water depth, and latitudinal (<100mm) or longitudinal (>100mm) coordinates of sample locations (not shown).

| Model number | Fixed effects | Model description |
|---|---|---|
| 1 | Intercept + Wet12 + Dry12 + Seascape + Estuary + Estuary × Wet12 + Estuary × Dry12 + Seascape × Wet12 + Seascape × Dry12 | Wet, normal, and dry 12-month flows each affect seascape- and estuary-scale variables differently |
| 2 | Intercept + Wet12 + Dry12 + Seascape + Estuary + Estuary × Wet12 + Estuary × Dry12 | Wet, normal, and dry 12-month flows each affect seascape- and estuary-scale variables and each affects the estuary -scale variable differently |
| 3 | Intercept + Wet12 + Dry12 + Seascape + Estuary + Seascape × Wet12 + Seascape × Dry12 | Wet, normal, and dry 12-month flows each affect seascape- and estuary-scale variables and each affects seascape-scale variables differently |
| 4 | Intercept + Wet6 + Dry6 + Seascape + Estuary + Estuary × Wet6 + Estuary × Dry6 + Seascape × Wet6 + Seascape × Dry6 | Wet, normal, and dry 6-month flows each affect seascape- and estuary-scale variables differently |
| 5 | Intercept + Wet6 + Dry6 + Seascape + Estuary + Estuary × Wet6 + Estuary × Dry6 | Wet, normal, and dry 6-month flows each affect seascape- and estuary-scale variables and each affects the estuary-scale variable differently |
| 6 | Intercept + Wet6 + Dry6 + Seascape + Estuary + Seascape × Wet6 + Seascape × Dry6 | Wet, normal, and dry 6-month flows each affect seascape- and estuary -scale variables and each affects seascape-scale variables differently |
| 7 | Intercept + Wet3 + Dry3 + Seascape + Estuary + Estuary × Wet3 + Estuary × Dry3 + Seascape × Wet3 + Seascape × Dry3 | Wet, normal, and dry 3-month flows each affect seascape- and estuary-scale variables differently |
| 8 | Intercept + Wet3 + Dry3 + Seascape + Estuary + Estuary × Wet3 + Estuary × Dry3 | Wet, normal, and dry 3-month flows each affect seascape- and estuary-scale variables and each affects the estuary-scale variable differently |
| 9 | Intercept + Wet3 + Dry3 + Seascape + Estuary + Seascape × Wet3 + Seascape × Dry3 | Wet, normal, and dry 3-month flows each affect seascape- and estuary-scale variables and each affects seascape-scale variables differently |
| 10 | Intercept + Wet1 + Dry1 + Seascape + Estuary + Estuary × Wet1 + Estuary × Dry1 + Seascape × Wet1 + Seascape × Dry1 | We, normal, and dry 1-month flows each affect seascape- and estuary-scale variables differently |
| 11 | Intercept + Wet1 + Dry1 + Seascape + Estuary + Estuary × Wet1 + Estuary × Dry1 | Wet, normal, and dry 1-month flows each affect seascape- and estuary-scale variables, and each affects the estuary-scale variable differently |
| 12 | Intercept + Wet1 + Dry1 + Seascape + Estuary + Seascape × Wet1 + Seascape × Dry1 | Wet, normal, and dry 1-month flows each affect seascape- and estuary-scale variables, and each affects seascape-scale variables differently |
| 13 | Intercept + Wet12 + Dry12 + Seascape + Estuary | Wet, normal, and dry 12-month flows each affect all seascape- and estuary-scale variables in the same way |
| 14 | Intercept + Wet6 + Dry6 + Seascape + Estuary | Wet, normal, and dry 6-month flows each affect all seascape- and estuary-scale variables in the same way |
| 15 | Intercept + Wet3 + Dry3 + Seascape + Estuary | Wet, normal, and dry 3-month flows each affect all seascape and estuary-scale variables in the same way |
| 16 | Intercept + Wet1 + Dry1 + Seascape + Estuary | Wet, normal, and dry 1-month flows each affect all seascape and estuary-scale variables in the same way |
| 17 | Intercept + Seascape + Estuary | No year-of-sampling flows effects are important |

probability for each size class; we selected the one with the lower $AIC_c$ for subsequent model fitting. To account for model selection uncertainty, we identified a confidence model set for each size class, defined as those models resulting in a cumulative Akaike weight of 0.95 (up to the model that resulted in equaling or surpassing an Akaike weight of 0.95; [47]). We assessed the precision of parameter estimates for each model in the confidence set by examining 95% confidence intervals and considered parameters important if their confidence interval did not contain zero. Lastly, we assessed goodness of fit for each model in the confidence set for each size class using a simulation-based approach to residual analysis, implemented in the R package "DHARMa" [48], to test for evidence of multi-collinearity (variance inflation factors), unexplained patterns in scaled residuals (QQ plots and quantile regressions), and spatial (Moran's I test) and temporal (Durbin-Watson test) autocorrelation.

**Model predictive performance.** We assessed in- and out-of-sample predictive performance for each model in the confidence set for each size class set using two approaches. First, we calculated an area under the receiving operator characteristic curve (AUC) statistic, which ranges from 0 to 1, where an AUC value greater than 0.5 indicates that a model predicts a given categorical outcome, on average, better than random chance alone [49]. Second, we calculated a Brier score, which ranges from 0 to 1, where, opposite AUC, a score closer to 0 indicates a better-predicting model [50]. Brier scores offer a slightly different (and complementary) measure of predictive performance than AUC because they explicitly incorporate the magnitude of predicted probabilities and can be interpreted as a measure of precision (i.e., mean squared error, where the square root of a Brier score is the average distance of predicted probabilities from the observed value, 1 or 0); in contrast, AUC values are based solely on rankings of predicted probabilities and not the magnitude of predicted probabilities. To assess the performance of each model, we split the data for each size class into separate training (80% of data) and testing data (20% of data) sets, re-fit the models in each size class's confidence set to the training data, and calculated testing and training AUC and Brier score statistics for both the training and testing data. To account for variability in model performance among randomly generated 20% testing and 80% training data sets, we repeated this process 1,000 times for each model in each size class's confidence set. For each size class and model, we then summarized the resulting sampling distribution of the AUC and Brier score statistics for both the testing and training data by calculating the mean, standard deviation, and coefficient of variation (%).

*Model predictions*. Prediction maps.—We calculated estuary-wide, model-averaged predictions of encounter probabilities for each of the four size classes using $AIC_c$ weights (re-scaled to sum to one across models in each confidence set) to weight the contribution of each model's predictions to the model-averaged predicted mean encounter probabilities. For these predictions, we used existing estuary-wide spatial layers (raster data used to create model covariates) for the seascape- and estuary-scale predictors. We exported the model predictions from R as GeoTiff files into a GIS [41]. In GIS, we clipped the spatial extent of prediction grids to near-shore shallow waters (depths $\leq$ ~1.8 m) that best-represented the extent of fish sampling locations. To demonstrate the influence of varying freshwater inflows on the predicted encounter probabilities, we calculated the model-averaged predictions for each size class under wet (the wettest observed freshwater inflows), normal, and dry (the driest observed freshwater inflows) hydrologic conditions.

Prediction line plots.–In addition to the estuary-wide predictions, we used line plots to demonstrate the influence of changing freshwater inflows and seascape- and estuary-scale variables by calculating mean encounter probabilities and 95% confidence intervals, again under three levels of freshwater inflow (wettest, normal, and driest observed conditions). Predictions for the line plots were calculated separately over the observed range of seagrass coverage (all

size classes), over the observed range of salt marsh coverage (only ≤50mm and 51–100mm size classes), over the observed range of distance to the nearest major inlet (all size classes), and over the observed range of habitat richness (all size classes). For simplicity, the line plot predictions used the top confidence set model in which each predictor variable occurred to make predictions (i.e., there were no model averaged predictions).

We conducted all analyses in R v4.0.3 [51] using the packages "glmmTMB" package (model fitting; [52]), "ROCR" (AUC; [53]), "rms" (Brier score; [54]), AICcmodavg (model selection; [55]), "raster" (raster stacking for predictions; [56]), and "biomod2" (model-averaged predictions;[57]).

## Results

The simulation-based goodness-of-fit assessments of all mixed effects logistic regression models in each size class's confidence set indicated no evidence of lack-of-fit based on visual assessments of QQ plots, quantile regressions of scaled residuals, and tests of scaled residuals for spatial and temporal autocorrelation. Predictive performance varied among models and size classes but was generally consistent among models for each size class and good to excellent for testing and training data sets, respectively, based on summaries of bootstrapped AUC and Brier scores (Table 2).

### 15–50mm size class

Model selection results indicated support for seven of the 17 candidate logistic regression models (Table 2). Based on $AIC_c$ weights, the best approximating model was at least 5.3 times more plausible than the second through seventh best-approximating models, and there was very little support for the remaining candidate models (Table 2). Among the confidence set of models, summed $AIC_c$ weights indicated the greatest support for 3-month freshwater inflow conditions interacting with seascape- and estuary-scale variables to influence seatrout encounter probabilities; however, there was some support for 1- and 6-month freshwater inflows as well. For seascape-scale variables, parameter estimates (see S1 Table for model-specific parameter estimates) from the confidence model set indicated that during normal freshwater inflow conditions, ≤50mm seatrout were generally more likely to be encountered in areas with extensive seagrass habitat (Table 3, Figs 3 and 4) and areas with higher habitat richness, and less likely to be encountered near oyster reefs. Under predominately wet conditions, seatrout were even more likely to be encountered in seagrass beds, whereas under predominately dry conditions only, seatrout were more likely to be encountered near salt marshes and less likely to be encountered in areas of high habitat richness. At the estuary scale, there was some evidence that under relatively recent (1 and 3-month flows), predominantly wet freshwater inflow conditions, seatrout ≤50mm were less likely to be encountered in areas farther away from major inlets relative to normal and dry conditions (Table 3, Fig 3); however, the relative support for these models was low (Models 8 and 11; Table 2 and S1 Table). Lastly, the parameter estimate associated with latitude indicated that seatrout were, on average, more likely to be encountered in the southern portions of the estuary. Parameter estimates for the remaining predictors were deemed unimportant as their 95% confidence intervals overlapped zero.

### 51–100mm size class

Model selection results indicated support for four of the 17 candidate logistic regression models (Table 2). Based on $AIC_c$ weights, the best approximating model was 1.7 and 2.1, and 7.5 times more plausible than the second, third, and fourth best-approximating models, respectively, and there was very little support for the remaining candidate models (Table 2). All

**Table 2. Model Number, number of parameters (K), AIC$_c$, ΔAIC$_c$, AIC$_c$ weights ($w$), cumulative AIC$_c$ weights ($wc$), and mean, standard deviation, and coefficient of variation (CV %) of bootstrapped testing and training data AUC and Brier score statistics from the confidence set of logistic regression models for each spotted seatrout size class.**

| Size Class and Model | K | AIC$_c$ | ΔAIC$_c$ | $w$ | $wc$ | Testing AUC | | | Training AUC | | | Testing Brier | | | Training Brier | | |
|---|---|---|---|---|---|---|---|---|---|---|---|---|---|---|---|---|---|
| | | | | | | Mean | SD | CV | Mean | SD | CV | Mean | SD | CV | Mean | SD | CV |
| *≤50mm* | | | | | | | | | | | | | | | | | |
| Mod9 | 19 | 2546.18 | 0.00 | 0.69 | 0.69 | 0.76 | 0.02 | 2.93 | 0.81 | 0.01 | 0.67 | 0.13 | 0.01 | 6.48 | 0.12 | 0.00 | 1.69 |
| Mod7 | 21 | 2549.58 | 3.40 | 0.13 | 0.81 | 0.76 | 0.02 | 2.95 | 0.81 | 0.01 | 0.67 | 0.13 | 0.01 | 6.48 | 0.12 | 0.00 | 1.70 |
| Mod8 | 13 | 2551.19 | 5.01 | 0.06 | 0.87 | 0.76 | 0.02 | 2.95 | 0.81 | 0.01 | 0.67 | 0.13 | 0.01 | 6.47 | 0.12 | 0.00 | 1.66 |
| Mod17 | 9 | 2552.38 | 6.20 | 0.03 | 0.90 | 0.76 | 0.02 | 2.94 | 0.81 | 0.01 | 0.67 | 0.13 | 0.01 | 6.49 | 0.12 | 0.00 | 1.66 |
| Mod11 | 13 | 2552.76 | 6.58 | 0.03 | 0.92 | 0.76 | 0.02 | 2.97 | 0.81 | 0.01 | 0.67 | 0.13 | 0.01 | 6.47 | 0.12 | 0.00 | 1.66 |
| Mod5 | 13 | 2553.60 | 7.42 | 0.02 | 0.94 | 0.76 | 0.02 | 2.95 | 0.81 | 0.01 | 0.67 | 0.13 | 0.01 | 6.48 | 0.12 | 0.00 | 1.67 |
| Mod14 | 11 | 2553.80 | 7.62 | 0.02 | 0.96 | 0.76 | 0.02 | 2.94 | 0.81 | 0.01 | 0.67 | 0.13 | 0.01 | 6.48 | 0.12 | 0.00 | 1.67 |
| *51–100mm* | | | | | | | | | | | | | | | | | |
| Mod9 | 19 | 1770.22 | 0.00 | 0.45 | 0.45 | 0.73 | 0.03 | 4.26 | 0.79 | 0.01 | 1.31 | 0.08 | 0.01 | 9.83 | 0.08 | 0.00 | 2.59 |
| Mod7 | 21 | 1771.29 | 1.07 | 0.27 | 0.72 | 0.73 | 0.03 | 4.23 | 0.79 | 0.01 | 1.31 | 0.08 | 0.01 | 9.81 | 0.08 | 0.00 | 2.60 |
| Mod8 | 13 | 1771.77 | 1.55 | 0.21 | 0.93 | 0.75 | 0.03 | 3.60 | 0.79 | 0.01 | 1.31 | 0.08 | 0.01 | 9.83 | 0.08 | 0.00 | 2.59 |
| Mod5 | 13 | 1774.18 | 3.96 | 0.06 | 0.99 | 0.73 | 0.03 | 4.26 | 0.79 | 0.01 | 1.31 | 0.08 | 0.01 | 9.87 | 0.08 | 0.00 | 2.57 |
| *101–150mm* | | | | | | | | | | | | | | | | | |
| Mod17 | 9 | 1703.40 | 0.00 | 0.42 | 0.42 | 0.75 | 0.03 | 3.69 | 0.82 | 0.01 | 0.89 | 0.15 | 0.01 | 6.67 | 0.14 | 0.00 | 2.03 |
| Mod5 | 13 | 1705.14 | 1.74 | 0.18 | 0.60 | 0.75 | 0.03 | 3.72 | 0.82 | 0.01 | 0.90 | 0.15 | 0.01 | 6.70 | 0.14 | 0.00 | 2.05 |
| Mod14 | 11 | 1705.72 | 2.32 | 0.13 | 0.73 | 0.75 | 0.03 | 3.68 | 0.81 | 0.01 | 0.90 | 0.15 | 0.01 | 6.66 | 0.14 | 0.00 | 2.03 |
| Mod16 | 11 | 1706.16 | 2.76 | 0.11 | 0.84 | 0.75 | 0.03 | 3.69 | 0.82 | 0.01 | 0.88 | 0.15 | 0.01 | 6.67 | 0.14 | 0.00 | 2.03 |
| Mod15 | 11 | 1707.38 | 3.98 | 0.06 | 0.90 | 0.75 | 0.03 | 3.69 | 0.82 | 0.01 | 0.89 | 0.15 | 0.01 | 6.67 | 0.14 | 0.00 | 2.03 |
| Mod13 | 11 | 1707.45 | 4.05 | 0.06 | 0.95 | 0.75 | 0.03 | 3.70 | 0.82 | 0.01 | 0.89 | 0.15 | 0.01 | 6.67 | 0.14 | 0.00 | 2.03 |
| *150–200mm* | | | | | | | | | | | | | | | | | |
| Mod17 | 9 | 1508.74 | 0.00 | 0.34 | 0.34 | 0.70 | 0.03 | 4.51 | 0.79 | 0.01 | 1.08 | 0.13 | 0.01 | 8.81 | 0.12 | 0.00 | 2.38 |
| Mod16 | 11 | 1510.28 | 1.54 | 0.16 | 0.49 | 0.70 | 0.03 | 4.60 | 0.79 | 0.01 | 1.12 | 0.13 | 0.01 | 8.82 | 0.12 | 0.00 | 2.38 |
| Mod14 | 11 | 1510.38 | 1.65 | 0.15 | 0.64 | 0.70 | 0.03 | 4.51 | 0.79 | 0.01 | 1.12 | 0.13 | 0.01 | 8.81 | 0.12 | 0.00 | 2.38 |
| Mod15 | 11 | 1510.74 | 2.00 | 0.12 | 0.77 | 0.70 | 0.03 | 4.51 | 0.79 | 0.01 | 1.10 | 0.13 | 0.01 | 8.79 | 0.12 | 0.00 | 2.38 |
| Mod5 | 13 | 1512.05 | 3.32 | 0.06 | 0.83 | 0.70 | 0.03 | 4.60 | 0.79 | 0.01 | 1.13 | 0.13 | 0.01 | 8.84 | 0.12 | 0.00 | 2.39 |
| Mod13 | 11 | 1512.37 | 3.63 | 0.05 | 0.88 | 0.70 | 0.03 | 4.50 | 0.79 | 0.01 | 1.08 | 0.13 | 0.01 | 8.81 | 0.12 | 0.00 | 2.38 |
| Mod8 | 13 | 1512.49 | 3.76 | 0.05 | 0.94 | 0.70 | 0.03 | 4.53 | 0.79 | 0.01 | 1.10 | 0.13 | 0.01 | 8.78 | 0.12 | 0.00 | 2.38 |
| Mod11 | 13 | 1513.41 | 4.67 | 0.03 | 0.97 | 0.70 | 0.03 | 4.63 | 0.79 | 0.01 | 1.11 | 0.13 | 0.01 | 8.83 | 0.12 | 0.00 | 2.39 |

**Table 3. Summary of the combined effects of freshwater inflow conditions, seascape-scale, and estuary-scale context variables based on parameter estimates from the confidence set of mixed effects logistic regression models for each seatrout size class.** (+) = positive effect; (-) = negative effect; (+/-) = inconclusive/no effect.

| Size Class | Seagrass coverage | Salt marsh coverage | Oyster bed coverage | Habitat richness | Distance to nearest major inlet |
|---|---|---|---|---|---|
| ≤50mm | (+) Normal/Dry; more (+) Wet | (+/-) Normal flows; (-) Wet; (+) Dry | (-) under all flow conditions | (+) Normal and Wet; (-) Dry | (+/-) Normal/Dry; (-) Wet |
| 51–100mm | (+) under all flow conditions | (-) Normal/Wet; (+) Dry | (+/-) under all flow conditions | (+) under all flow conditions | (+/-) Normal/Dry; (-) Wet |
| 101–150mm | (+) under all flow conditions | (+/-) under all flow conditions | (+/-) under all flow conditions | (+) under all flow conditions | (+) Normal/Wet; more (+) Dry |
| 151–200mm | (+) under all flow conditions | (+/-) under all flow conditions | (-) under all flow conditions | (+) under all flow conditions | (+) under all flow conditions |

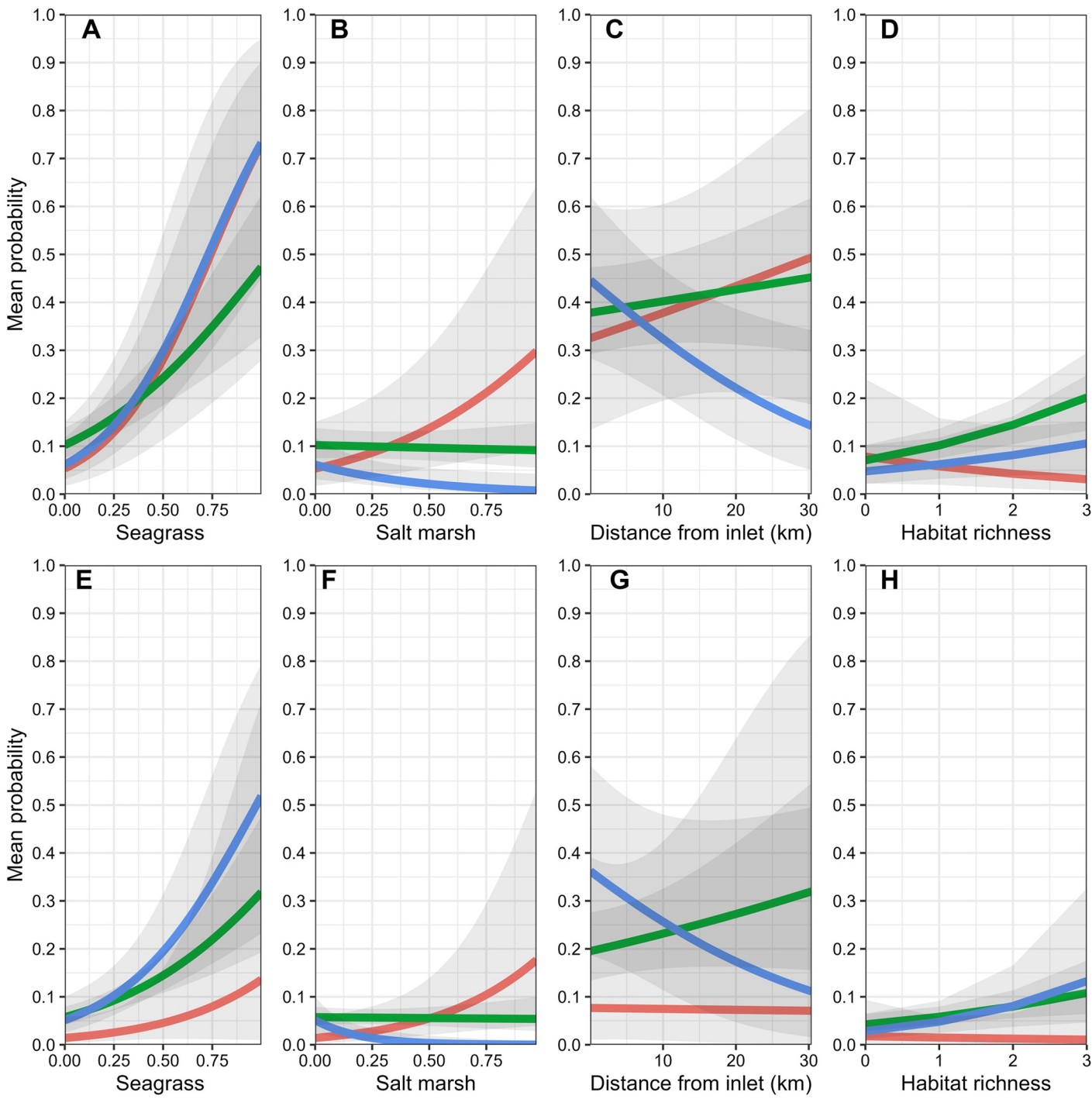

**Fig 3.**

models in the confidence model set included various combinations of 3- and 6-month wet and dry freshwater inflow variables interacting with seascape- and estuary-scale variables; however, the bulk of support based on $AIC_c$ weights was for a model that included 3-month freshwater inflows interacting with seascape-scale variables. For seascape-scale variables, parameter

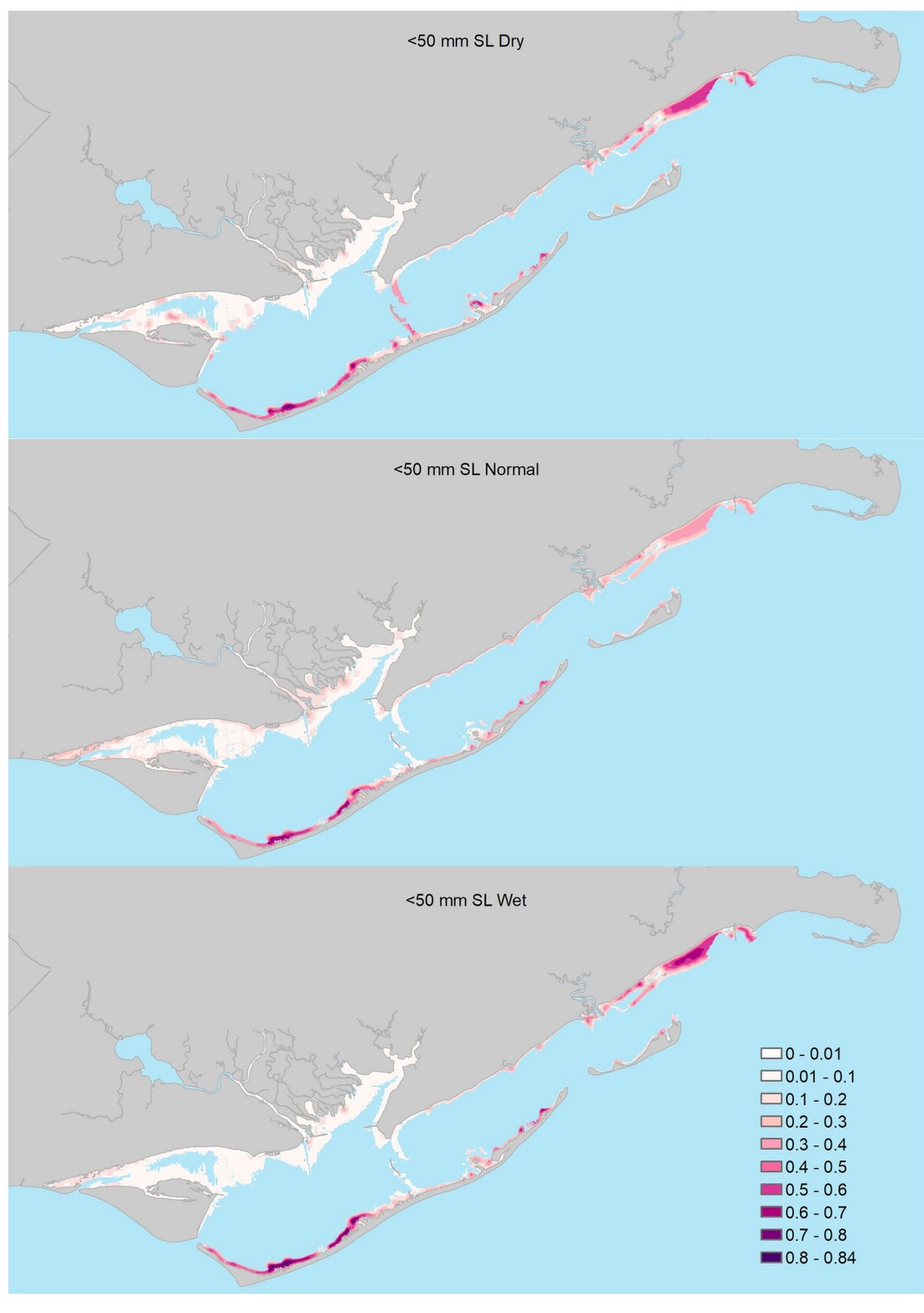

**Fig 4.**

estimates (see S2 Table for model-specific parameter estimates) from the confidence model set indicated that under predominantly normal monthly freshwater inflow conditions, 51–100mm seatrout were more likely to be encountered in seagrass beds (Figs 3 and 5) and areas of high habitat richness, and less likely to be encountered near salt marshes (Table 3). Under predominantly wet conditions within 3-months of peak recruitment, 51–100mm seatrout were less likely to be encountered near salt marshes, whereas they were more likely to be encountered near salt marshes under predominately dry conditions. Additionally, modeling results also suggested that predominantly wet conditions within 6-months prior to peak recruitment increased encounter probabilities of 51–100mm seatrout in all seascape-scale habitats (i.e., there was no evidence that this effect differed among seascape-scale habitats; Fig 5). At the estuary-scale (see S2 Table), similar to ≤50mm seatrout, 51–100mm seatrout were less likely to be encountered in western bay under recent (3-month flows) wet freshwater inflow conditions (Table 2 and S2 Table). Lastly, 51–100mm seatrout were, in general, more likely to be encountered in the southern portions of the estuary. Parameter estimates for the remaining predictors were deemed unimportant as their 95% confidence intervals overlapped zero.

### 101–150mm size class

Model selection results indicated support for six of the 17 candidate logistic regression models (Table 2). Based on $AIC_c$ weights, the best approximating model was at least 2.3 times more plausible than the second though sixth best-approximating models (Table 2). The best-approximating model did not include any freshwater inflow variables, although there was some support for models that included the influence of freshwater inflows at 1 (month of sampling), 3, 6, and 12-month windows. Only one model in the confidence set included interactions between freshwater inflow variables and the estuary-scale variable, and no models included freshwater inflow variables interacting with seascape-scale variables (Table 2). For seascape-scale habitat variables, parameter estimates (see S3 Table) from the confidence model set indicated that under all freshwater inflow conditions, 101–150mm seatrout were more likely to be encountered in areas with relatively abundant seagrass beds (Table 3, Figs 6 and 7), and high habitat richness, and deeper water, and less likely to be encountered near oyster reefs (Table 3). At the estuary scale (see S3 Table for model-specific parameter estimates), 101–150mm seatrout were more likely to be encountered farther from major inlets, particularly during predominantly dry conditions within 6-months of peak recruitment (Table 2 and S3 Table; Fig 6). Lastly, the parameter estimate associated with longitude indicated that 101–151mm seatrout were, in general, more likely to be encountered in western portions of the estuary. Parameter estimates for the remaining predictors were deemed unimportant as their 95% confidence intervals overlapped zero.

### 151–200mm size class

Model selection results indicated support for eight of the 17 candidate logistic regression models (Table 2). Based on $AIC_c$ weights, the best approximating model was at least 2.1 times more plausible than the second through eighth best-approximating models, and there was very little support for the remaining candidate models (Table 2). Similar to the 101–150mm size class, the best-approximating model did not include any freshwater inflow variables, but there was some support for models that included the influence of freshwater inflows at 1 (month of sampling), 3, 6, and 12-month windows. However, none of the parameter estimates associated with freshwater inflows in these models were deemed important as their 95% confidence intervals overlapped zero. Four of the eight models in the confidence set included interactions between freshwater inflow variables and the estuary-scale variable, and no models included

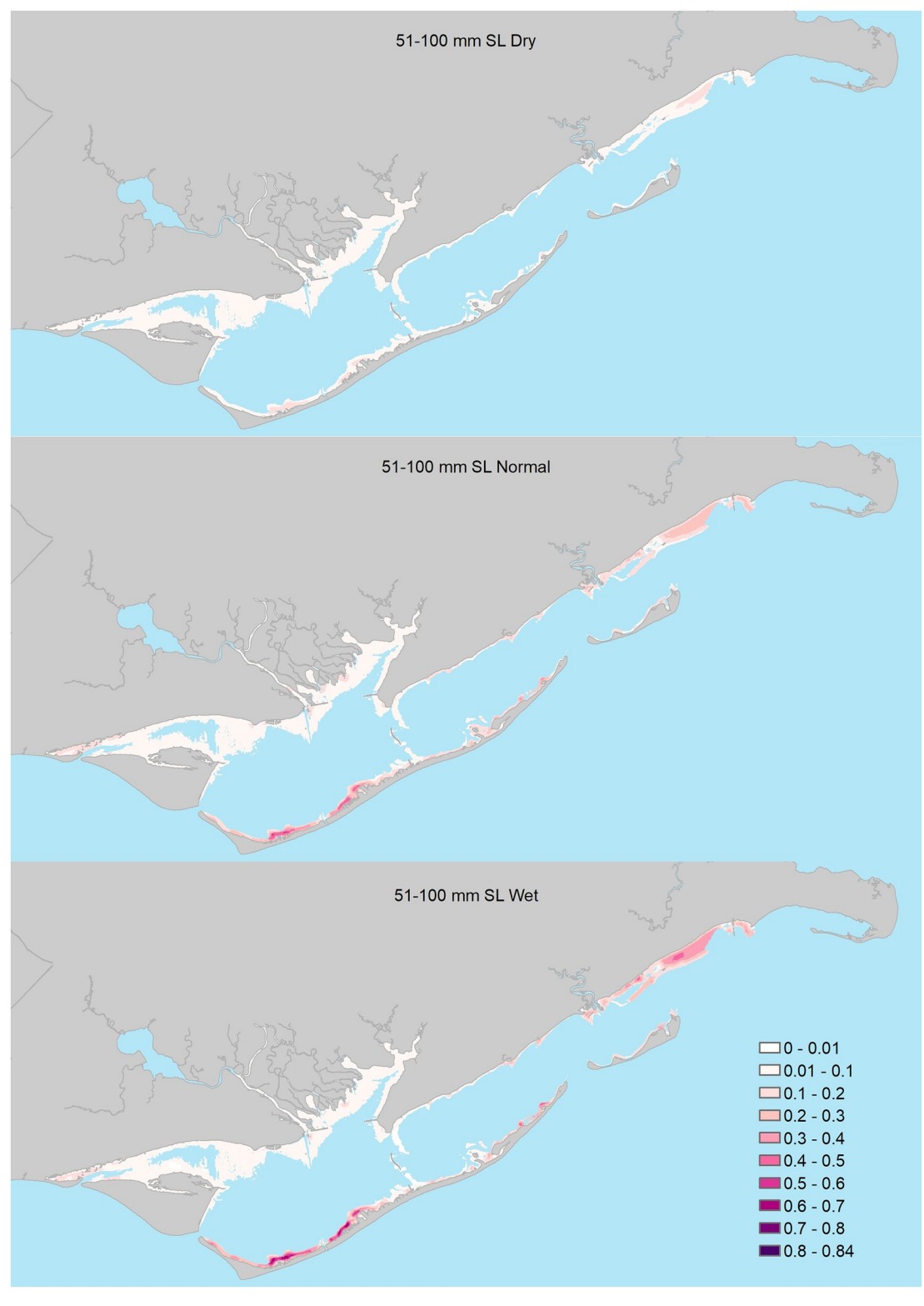

**Fig 5.**

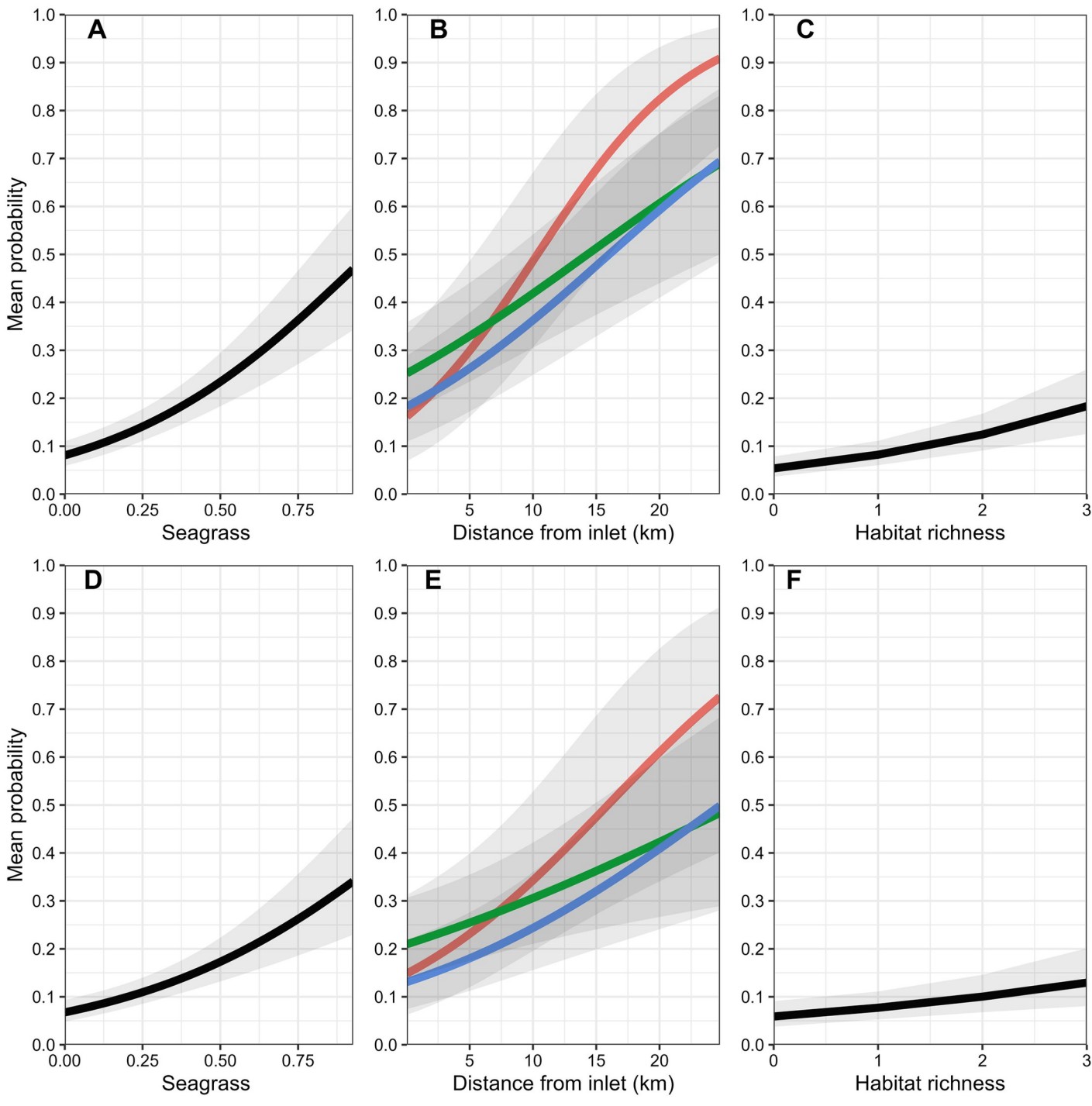

**Fig 6.**

freshwater inflow variables interacting with seascape-scale habitat variables (Table 2). For seascape-scale habitat variables, parameter estimates (see S4 Table for model-specific parameter estimates) from the confidence model set indicated that under all freshwater inflow conditions, 151–200mm seatrout were more likely to be encountered in areas with abundant seagrass beds

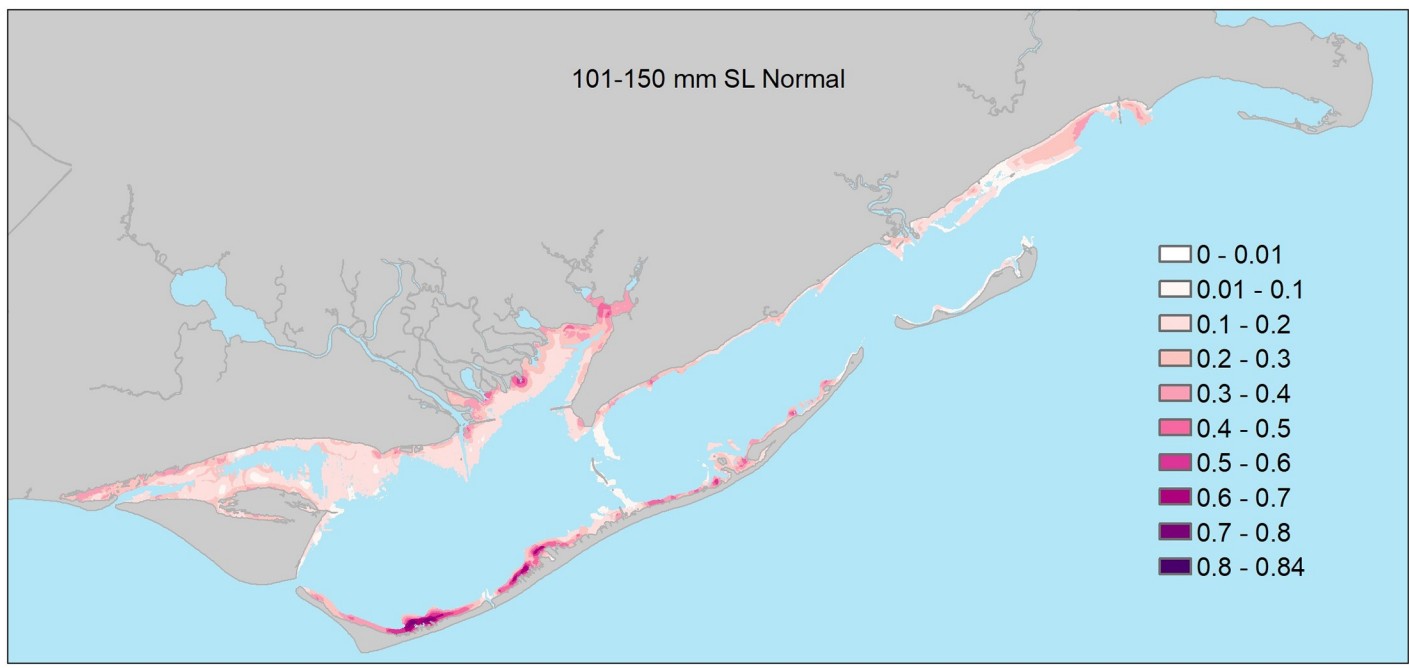

**Fig 7.**

(Table 3, Figs 6 and 8) and high habitat richness within a 400-m radius, and deeper water. At the estuary scale, 151–200mm seatrout were more likely to be encountered farther from major inlets during all freshwater inflow conditions (Table 3, Fig 6). Lastly, the parameter estimate associated with longitude indicated that 151–200mm seatrout were, in general, more likely to

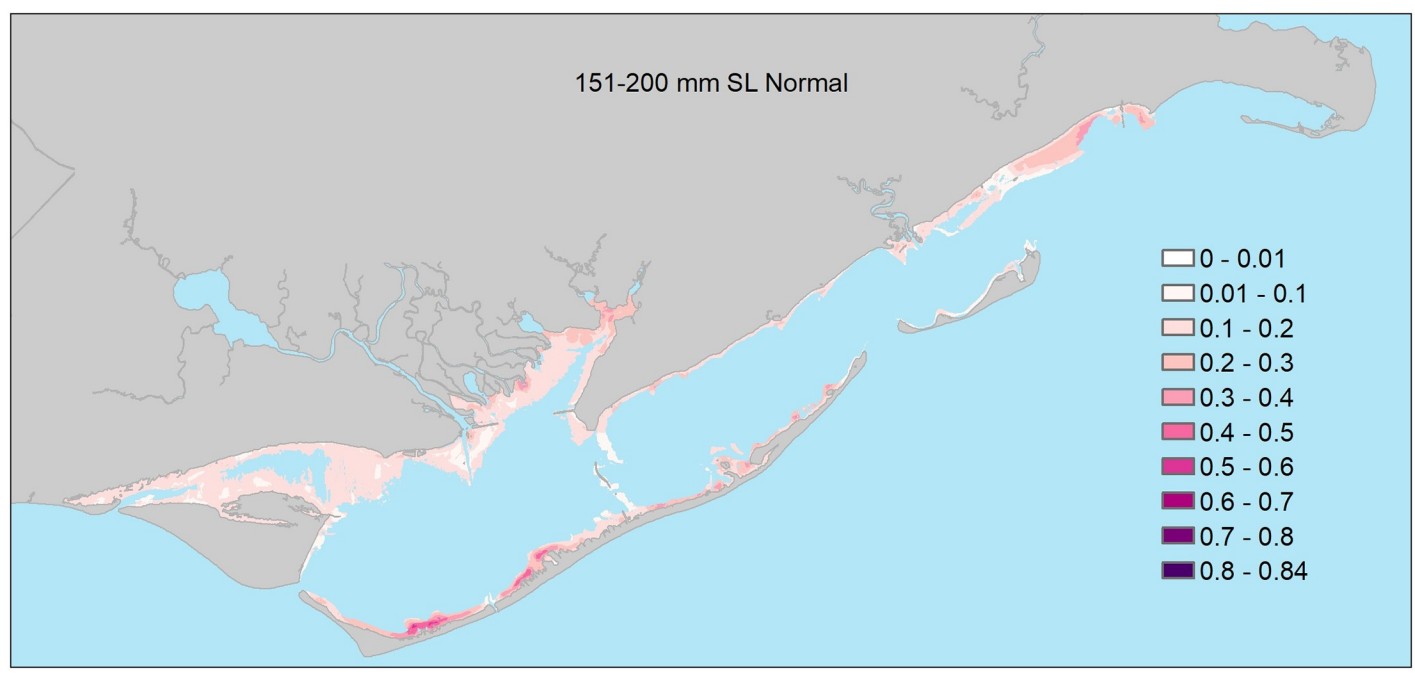

**Fig 8.**

be encountered in the western portions of the estuary. Parameter estimates for the remaining predictors were deemed unimportant as their 95% confidence intervals overlapped zero.

## Discussion

The present study revealed a consistent, positive influence of seagrass area (within a 400-m radius of fish collection sites) on the encounter probability of all four size classes (15mm–200mm SL) of seatrout, regardless of inflow conditions, confirming the importance of this habitat, particularly large areas of seagrass, on juvenile stages of this economically-important fishery species in Apalachicola Bay. In addition, habitat richness within the same 400-m radius was also positively related to encounter probability of all size classes of juvenile seatrout except the smallest size class during dry inflow periods. The smallest juveniles were even more closely associated with seagrass habitat during dry conditions, and also occurred more frequently in areas with nearby salt marshes.

The models showed that the encounter probability of the two smaller sizes (15–50mm and 51–100mm SL) were most closely related to freshwater inflow conditions within the three months leading up to and including peak recruitment in Apalachicola Bay compared with shorter (1 month) and longer (6 and 12 month) periods. The best models describing occurrence patterns of the two smaller size classes contained the same exact set of predictors from two datasets representing each size class providing strong evidence for the models in general, as well as the importance of inflow during these early life stages. Models for encounter probability of the larger juveniles (101–150mm and 151–200mm SL) showed they were both generally unrelated to inflow conditions but were more likely to be encountered in areas containing abundant seagrass and multiple nearby habitat types.

### Seascape-scale context

**Area of biogenic habitat types.**  Although juveniles of many estuarine-dependent species are associated with estuarine biogenic habitats, species-habitat relationships have largely been examined by characterizing the presence or absence of habitat types observed within sample sites [6]. The influence of habitat proximity on the taxonomic composition of estuarine nekton assemblages is well established [58–65]. However, the present study is one of the first to examine relationships between juvenile seatrout occurrence and seascape context (but see [15]). Juvenile seatrout are often associated with seagrass habitat in many estuaries [20, 22, 24, 27, 30]. The positive relationships we found between seatrout occurrence (all four size classes) and area of seagrass in Apalachicola Bay may be a result of increased benthic prey typically found in these habitats as well as enhanced protection from predation compared with unvegetated areas (see [66] for review). Larger, continuous patches of seagrass may allow the juveniles to feed over a larger habitat area on benthic resources which are typically relatively abundant in seagrass while remaining protected from predators thereby reducing the need to move into or through unvegetated areas [67]. Relatively large areas of seagrass habitat measured at broad spatial scales has recently been related to relatively high densities of benthic feeders within nekton communities in other estuaries [64, 67], suggesting that increased benthic resources within large areas of seagrass habitat may be responsible for observed habitat use patterns in juvenile seatrout as they grow and transition from plankton to benthic food sources [20].

**The influence of habitat richness.**  Many estuarine species, including juveniles of fishery species, regularly move among nearby intertidal and subtidal habitat types to forage and seek refuge from predation [10]. This study is one of the first to report positive relationships between juvenile seatrout occurrence and the number of habitat types (habitat richness) within a local area (400-m radius) around the fish collection sites. All four size classes of juveniles,

with one exception of the smallest juveniles during dry conditions, were more likely to be encountered in areas with relatively high habitat richness regardless of freshwater inflow conditions. Similarly, catch rates of juvenile seatrout as well as other fishery species, such as penaeid shrimp *Farfantepenaeus spp*. and gray snapper *Lutjanus griseus* were higher in areas with nearby saltmarsh and seagrass habitat compared with areas with only one of these habitats in a North Carolina estuary [9]. Larger-sized adolescent seatrout (240–308mm) were also found to reside preferentially in seascapes containing multiple habitat types within a Texas estuary [68]. Relatively high abundance of many other common estuarine species, such as pinfish *Lagodon rhomboides* and blue crabs *Callinectes sapidus*, have also been related to the high connectivity of multiple nearby habitats such as seagrass, oysters, and saltmarsh [69, 70]. Similarly, seascape context was important in determining habitat function for juvenile fishes in an Australian estuary [12]. Our results on juvenile seatrout add to the growing number of studies showing that seascape context is important to habitat use patterns of estuarine-dependent fishery species [6, 12–14, 17].

### Influence of freshwater inflow on early juvenile seatrout

Juvenile stages of spotted seatrout respond to changes in freshwater inflow in estuaries along Florida's Gulf Coast [28, 30, 32]. Yet, the relative influence of freshwater inflow on different life stages of seatrout in Apalachicola Bay is not clear. We found that freshwater inflow conditions in our study during the three months prior to and including peak recruitment were the most closely related to encounter probability of the earliest life stages (15–100mm SL) compared with longer (6- and 12-month) and shorter (1-month) time periods. The heaviest recruitment of smaller juveniles (15–50mm and 51–100mm SL size classes) occurred from July through September, suggesting that freshwater inflow conditions during May through September are particularly influential on juvenile recruitment of spotted seatrout. In contrast, river flow from March to May explained most of the variation in juvenile seatrout abundance (≤100mm SL) in the nearby Suwannee River Estuary located 185 km southeast of Apalachicola Bay [32]. Unlike smaller juveniles, we found that occurrence of larger juvenile size classes (101–200mm SL) in Apalachicola Bay were unrelated to inflow conditions, regardless of the time period considered. Interestingly, the influence of freshwater inflow on the relative abundance of juvenile seatrout in Tampa Bay also diminished as the juveniles reached the 51–100mm SL size class [28]. The reduced influence of freshwater inflow on these later juvenile stages may reflect ontogenetic changes in habitat use or diet as juveniles settle out of the plankton into benthic habitats in Apalachicola Bay.

The timing of this 3-month period coincides with the egg, larval, and early juvenile stages of seatrout and therefore suggests that inflow may influence seatrout survival and growth in the earliest life stages. The peak of spawning occurs within this 3-month period [71]; however, it seems unlikely that reproductive success is influenced by freshwater inflow because seatrout appear to be locally adapted to the prevailing salinity regime within their home estuary [19] and spawning as well as eggs and early larvae occur in a wide range of salinities [72, 73]. However, rapidly changing salinity conditions may cause osmoregulatory stress on larvae and juveniles [74, 75]. In addition, inflow conditions within an individual estuary may affect planktonic egg and larval stages by influencing transport processes, planktonic food availability and perhaps predation pressure. After spawning, eggs and larvae are planktonic for up to 17 days prior to benthic settlement [20]. During this stage, high levels of freshwater inflow may push planktonic eggs and seatrout larvae seaward, potentially altering the location of their benthic settlement as seen in other species [76–79]. We found weak evidence that during wet conditions, the two smaller size classes were less likely to occur in areas located farther from

the nearest inlet to the Gulf of Mexico suggesting that heavy freshwater inflow may have flushed these smaller fish out toward the inlet. These areas may have also undergone a reduction in salinity with increased inflow during wet conditions that could have caused osmoregulatory stress to the early life stages of seatrout.

In addition to potential changes in transport processes and salinity fluctuations, periods of relatively high inflow typically coincide with increased nutrient levels, elevated turbidity, as well as increased inundation of saltmarshes within the riverine floodplains. Increased nutrients can lead to a boost in plankton [3, 80, 81] an essential food source for larval (3–4.5mm SL, [75]) and early juvenile seatrout (15–30mm SL, [20]). This increase in planktonic food may play a role in increasing survivability and therefore the probability of encountering early juveniles during relatively high inflow conditions. Smaller seatrout (15–30mm SL) consumed planktonic organisms, but larger juveniles (31–100mm SL) transitioned into a benthic diet in Tampa Bay [20]. Therefore, larger juveniles may not be affected by fluctuations in plankton as they have shifted their diet from plankton to more benthic sources which are not as closely tied to short-term changes in freshwater inflow and associated nutrient levels [81]. Periods of high freshwater inflow can also elevate turbidity throughout the estuary which may also lower predation pressure on larvae and early juveniles by reducing predator visibility [2, 3]. In fact, river flow reduction in Apalachicola Bay during a 2-yr drought resulted in reduced nutrient loading and turbidity which triggered a change in the overall trophic structure of the bay [82]. Finally, heavy freshwater inflow may increase flooding of salt marshes thus allowing juveniles greater access to abundant benthic prey typically available in these habitats [2–3]. In this study, increased flooding also likely caused juvenile seatrout within inundated salt marshes to be less susceptible to seine capture in wet periods causing some counterintuitive results. Although the relative importance of mechanisms remains unclear, early life stages of seatrout appear to be the most vulnerable to reductions in inflow in Apalachicola Bay.

## Differences in biogenic habitat use patterns in relation to inflow conditions

Although habitat use patterns for most size classes of juvenile seatrout were generally consistent regardless of inflow conditions, biogenic habitat use changed dramatically for the smallest size (15–50mm SL) of seatrout during dry inflow conditions. During normal and wet inflow conditions, these small juveniles were often encountered in areas characterized by relatively large amounts of seagrass, high habitat richness, and limited salt marsh habitat. During wet conditions, small juveniles were encountered even more frequently in seagrass habitat, particularly in areas approaching 100% coverage; the dominance of seagrass precluded other habitat types and resulted in relatively low overall habitat richness in these areas. These results suggest that subtidal seagrass habitats may be more important to early juvenile seatrout in wet conditions. We also found that the two smallest size classes of juveniles were encountered more frequently in areas with nearby salt marsh habitat during dry conditions but less frequently during normal and wet conditions. Salt marsh habitat generally occurred in areas farther away from the inlet and we did find some weak evidence of a broad-scale spatial shift in encounter probability relative to the inlet during wet conditions. Therefore, small seatrout may have been either flushed downstream from marsh areas during wet conditions or responded to unfavorable salinity changes in these areas. Alternatively, the observations of greater encounter probability near marshes during dry conditions may be related to a limitation of the seine gear used in this study. Juvenile seatrout occur in relatively high probability along salt marsh shorelines in northern Gulf of Mexico estuaries [21, 83], and may also utilize the salt marsh interior (away from the edge). During wet and normal conditions, the intertidal salt marsh surface is likely inundated more frequently and thus more available for use by fish than during dry

conditions [3]. However, the seine was deployed along the marsh edge only, and was not able to capture seatrout potentially using the interior marsh surface, particularly during high inflow periods. During dry conditions when marshes were less inundated, fish were likely pushed into nearby subtidal areas and more accessible to seine capture. Had the salt marsh interior been sampled directly, the area of salt marsh may have been more related to encounter probability of juvenile seatrout. Overall, habitat use changes during dry conditions further demonstrate the influence of freshwater inflow on these early juvenile seatrout.

## Implications for management and future research

This study provides insight into the importance of seascape context and freshwater inflow on juvenile stages of seatrout. Juvenile seatrout (all size classes) were most frequently encountered in seascapes containing relatively large areas of seagrass as well as additional habitat types, such as saltmarshes and oyster reefs nearby. Based on habitat maps used in this study, these types of seascapes are currently relatively abundant in Apalachicola Bay as it is one of the less developed estuaries in Florida. However, as human populations grow, development may threaten these areas by either direct development of shorelines as has occurred in many Florida estuaries, or indirectly by degradation of water quality known to negatively impact seagrass habitat [84]. Identification and mapping of these seascapes as important nursery areas for seatrout is the first step in developing conservation strategies to protect them as human development may increase in this area. Additionally, this information can be used to design and restore mosaics of functionally connected habitats to improve nursery seascape habitat for fishery species such as seatrout [85]. Future research could expand the seascape approach into other estuaries within the range of seatrout to compare habitat use patterns found in Apalachicola Bay to seascapes used elsewhere. For example, juvenile seatrout used marsh habitat in areas with limited seagrass in several northern Gulf of Mexico estuaries [27]. Estuaries will likely contain seascapes that are at least somewhat unique based on the context of local geomorphology as well as the relative availability of seagrass and other structurally complex habitat types [16, 86]. In addition to habitat use, relationships between inflow conditions and encounter rates of juvenile seatrout as described may also help develop water management strategies to potentially assist recovery of declining seatrout populations, particularly targeting freshwater inflow during the period prior to and including peak recruitment. The human need for freshwater likely will increase. Therefore, we need a better understanding of how further reductions of freshwater inflow may adversely affect fishery species with significant economic and cultural importance to local communities. These results add to a growing body of literature aimed at understanding the influence of freshwater inflow on vulnerable juvenile life stages of fishery species to provide more informed strategies for freshwater inflow management.

## Supporting information

**S1 Table. Models for ≤50 mm spotted seatrout.** Parameter estimates, standard errors, lower and upper 95% confidence limits, and Wald z-scores (*z*) and p-values (*p*) from the confidence set of mixed effects logistic regression models relating seascape-scale, estuary-scale, and hydrologic variables to the probability of encountering ≤ 50 mm spotted seatrout. All values are on the logit (log-odds) scale, random effects are reported as standard deviations, and Imp denotes statistically important relationships based on an alpha level of 0.05.
(DOCX)

**S2 Table. Models for 51–100 mm spotted seatrout.** Parameter estimates, standard errors, lower and upper 95% confidence limits, and Wald z-scores (*z*) and p-values (*p*) from the confidence set of mixed effects logistic regression models relating seascape-scale, estuary-scale, and hydrologic variables to the probability of encountering 51–100 mm spotted seatrout. All values are on the logit (log-odds) scale, random effects are reported as standard deviations, and Imp denotes statistically important relationships based on an alpha level of 0.05.
(DOCX)

**S3 Table. Models for 101–150 mm spotted seatrout.** Parameter estimates, standard errors, lower and upper 95% confidence limits, and Wald z-scores (*z*) and p-values (*p*) from the confidence set of mixed effects logistic regression models relating seascape-scale, estuary-scale, and hydrologic variables to the probability of encountering 101–150 mm spotted seatrout. All values are on the logit (log-odds) scale, random effects are reported as standard deviations, and Imp denotes statistically important relationships based on an alpha level of 0.05.
(DOCX)

**S4 Table. Models for 151–200 mm spotted seatrout.** Parameter estimates, standard errors, lower and upper 95% confidence limits, and Wald z-scores (*z*) and p-values (*p*) from the confidence set of mixed effects logistic regression models relating seascape-scale, estuary-scale, and hydrologic variables to the probability of encountering 151–200 mm spotted seatrout. All values are on the logit (log-odds) scale, random effects are reported as standard deviations, and Imp denotes statistically important relationships based on an alpha level of 0.05.
(DOCX)

## Acknowledgments

We are grateful to the Florida Fish and Wildlife Research Institute's Fisheries Independent Monitoring program (Apalachicola lab) for collecting the field data used in this analysis. In addition, we greatly appreciate support by senior staff (R. Baumstark, R. Flamm, and L. McEachron) and the entire Center for Spatial Analysis team at FWRI. We appreciate Philip Stevens, Paul Schueller, Richard Flamm, Rene Baumstark, Mariah Livernois, and two anonymous reviewers for providing helpful input on earlier drafts of the manuscript.

## Author Contributions

**Conceptualization:** Shannon D. Whaley, Colin P. Shea, David A. Gandy.

**Data curation:** Shannon D. Whaley, Colin P. Shea, E. Christine Santi, David A. Gandy.

**Formal analysis:** Shannon D. Whaley, Colin P. Shea, E. Christine Santi, David A. Gandy.

**Funding acquisition:** Shannon D. Whaley, David A. Gandy.

**Investigation:** Shannon D. Whaley, Colin P. Shea, David A. Gandy.

**Methodology:** Shannon D. Whaley, Colin P. Shea, E. Christine Santi, David A. Gandy.

**Project administration:** Shannon D. Whaley.

**Resources:** Shannon D. Whaley, E. Christine Santi, David A. Gandy.

**Software:** Colin P. Shea, E. Christine Santi.

**Supervision:** Shannon D. Whaley, David A. Gandy.

**Validation:** Colin P. Shea, David A. Gandy.

**Visualization:** E. Christine Santi.

**Writing – original draft:** Shannon D. Whaley, Colin P. Shea, E. Christine Santi, David A. Gandy.

**Writing – review & editing:** Shannon D. Whaley, Colin P. Shea, E. Christine Santi, David A. Gandy.

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
