## [Decision Letter · Decision Letter 0]

16 Aug 2023

PONE-D-23-22016The influence of freshwater inflow and seascape context on occurrence of juvenile Spotted seatrout Cynoscion nebulosus across a temperate estuaryPLOS ONE

Dear Dr. Whaley,

Thank you for submitting your manuscript to PLOS ONE. After careful consideration, we feel that it has merit but does not fully meet PLOS ONE’s publication criteria as it currently stands. Therefore, we invite you to submit a revised version of the manuscript that addresses the points raised during the review process.

We look forward to receiving your revised manuscript.

Kind regards,

Masami Fujiwara, PhD

Academic Editor

PLOS ONE

Journal Requirements:

3. We note that Figures 1,2,4,5,7 and 8 in your submission contain map/satellite images which may be copyrighted. All PLOS content is published under the Creative Commons Attribution License (CC BY 4.0), which means that the manuscript, images, and Supporting Information files will be freely available online, and any third party is permitted to access, download, copy, distribute, and use these materials in any way, even commercially, with proper attribution. For these reasons, we cannot publish previously copyrighted maps or satellite images created using proprietary data, such as Google software (Google Maps, Street View, and Earth). For more information, see our copyright guidelines: http://journals.plos.org/plosone/s/licenses-and-copyright.

a. You may seek permission from the original copyright holder of Figures 1,2,4,5,7 and 8 to publish the content specifically under the CC BY 4.0 license.  

Additional Editor Comments:

We received comments from three reviewers, who are experts on the subjects (species, ecology, and/or methodology). All of the comments are very positive. It is consistent with my assessment as well. I agree with Reviewer 1 that some clarifications are needed. From my experience, when clarifications are needed, we might discover some problems after revisions. For now, I expect minor revisions for clarifications are sufficient. However, I will read a revised version carefully and if necessary, I will send it for reviews again.  

Reviewers' comments:

Reviewer's Responses to Questions

**Comments to the Author**

1. Is the manuscript technically sound, and do the data support the conclusions?

Reviewer #1: Partly

Reviewer #2: Yes

Reviewer #3: Yes

2. Has the statistical analysis been performed appropriately and rigorously? 

Reviewer #1: Yes

Reviewer #2: Yes

Reviewer #3: Yes

3. Have the authors made all data underlying the findings in their manuscript fully available?

Reviewer #1: Yes

Reviewer #2: Yes

Reviewer #3: No

4. Is the manuscript presented in an intelligible fashion and written in standard English?

Reviewer #1: Yes

Reviewer #2: Yes

Reviewer #3: Yes

5. Review Comments to the Author

Reviewer #1: This manuscript describes the effects of freshwater inflow on the habitat use patterns of juvenile spotted seatrout across four distinct life stages. The authors found that habitat use patterns were influenced by freshwater inflow (primarily during the 3-months prior to recruitment) for the smallest size classes, but less so for the larger juveniles. I found the methods to be sound, but encountered some confusion about how the freshwater inflow variables were determined and incorporated into the models. The time lag component was a bit unclear, and didn’t come across well in the figures. The authors also make some statements and conclusions regarding the effect of freshwater inflow on presence probability, but from what I understand the results only show the effect of freshwater inflow on spatial distribution of seatrout and their use of biogenic habitats. This wording needs to be clarified throughout the manuscript and possibly visually clarified via figures. The figures could benefit from some reworking, since I found that they didn’t clearly visualize some of the patterns the authors discuss (especially habitat richness). I also wonder if more detailed figures would help alleviate confusion regarding the freshwater inflow variables. I am supportive of this manuscript for publishing following revisions. Specific comments below:

Abstract:

- Line 23: suggest “recreational sportfish”

- Line 26: “We used”

- Lines 34-35: I'm not sure how we can determine a positive relationship between presence and inflow based on the figures provided. From what I can gather based on the figures, the models assess the effect of inflow on habitat use patterns, not the effect of inflow on presence probability. Reconsider this wording.

- Line 43: suggest changing “larger sized fishes” to “larger juveniles”

Introduction:

- Lines 57-63: Could be informative to add some text here linking freshwater inflow patterns with these habitats. Habitat-forming plants and animals also respond to changes in freshwater inflow, meaning the two concepts (freshwater flow/salinity and availability of structured habitat) are inherently linked. Furthermore, the results suggest use of different habitats depends on inflow conditions, so the two ideas should be more cohesive instead of separate (as written).

- Lines 63-65: With the addition of text above, this could become a new paragraph about habitat richness.

- Line 79: “popular recreational fishery species”

- Line 80: “live out life” is odd wording, suggest something like “spend their lifespan” or “are resident within”

- Line 85: I suggest staying consistent as possible with “juvenile spotted seatrout” throughout the manuscript. (Here it says juveniles of seatrout, but later says juvenile spotted seatrout).

- Line 120-121: Maybe give context here for peak recruitment – which months/time of year?

Methods:

- Lines 177-184: Has there been any change in saltmarsh/seagrass/oyster coverage across the study period (2001-2018), and do you think the coverage you have is reasonably close to the coverage across that timescale? I would suggest adding a justification for the mismatch between habitat and fish data. I appreciate the challenges with getting high res habitat data, so I assume you did the best you could, but a quick discussion of potential caveats would be good to include.

- Line 187: Add a little more detail about QA/QC – not sure what standard means

- Lines 194-198: Could do this same calculation for distance from the major freshwater inflow source. Looking at Fig 1 there are collection points way out East in the bay that are far from the river, and others that are immediately adjacent to the river. I would expect that those locations would not experience the same magnitude of influence of inflow, right? Giving all locations the same inflow value seems like you might be missing some important spatial patterns of inflow effect.

- Lines 206-208: This sentence is confusing - please clarify wording and provide a more specific citation

- Lines 209-211: Would be helpful to add a marker for the stream gauge location on Fig 2 to give spatial context

- Lines 211-212: This is essentially a time lag, right? This determines whether recent or past conditions affect the fish more or less? Please clarify the wording of this.

- Lines 213-215: How is this incorporated into the analyses? I'm having a hard time understanding what the freshwater inflow predictor variables actually look like. I would think that for each seine data point, there would be a classification of wet, dry, or normal during the month prior, and 3, 6, and 12 months prior - is that correct? I think there needs to be a clearer explanation of how these variables truly operate within the models.

- Lines 272-273: This wording is confusing - the sentences above make it sound like AUC and Brier scores were calculated for each model (without testing vs. training datasets). Consider changing the order of how this process is presented, starting with the splitting of data into testing and training, followed by AUC and Brier score calculations, and ending with how you summarized the performance results.

- Lines 282-283: Does this refer to calculated predictions of encounter probabilities across the range of values for each predictor variable? The wording in this section is hard to follow

- Lines 285-287: Don't the models already include values for the habitat variables around each sample location? Wouldn't the range of those values be used for prediction? This makes it sound like it’s predicting spatially, but I'm not sure why that would be necessary here (since it occurs in the prediction maps section)

- Lines 287-290: I assume this is what we're seeing in Figs 3 and 6, but it isn't super clear. Predicted encounter probabilities were calculated across the possible range of each predictor variable for each inflow condition, correct?

- Line 293: How does the above paragraph describing model prediction link to these maps. I see here that you've exported a spatial map of predictions, but then it's later stated (last sentence of this paragraph) that you only used the top model for this. Aren't the predictions coming from the weighted/averaged models above? Please clarify this section and the previous paragraph to clearly state what models and predictions relate to each figure (effects vs. maps)

Results:

- Line 329: This isn't something I can easily distinguish from the figures - There's no figure that shows the relationship between habitat richness and spotted seatrout presence

- Line 330: Same here, no visualization of the oyster reef trend

- Lines 330-332: I suggest adding figure labels (including panel labels) here to distinguish exactly what fig the reader should look at to see these patterns

- Line 334: What does prolonged mean in this context? This word is used throughout the results but in relation to all time lags (1,3,6,12 months), which doesn't make sense to me. Define what "prolonged" means in the context of the freshwater flow variables

Discussion

- Lines 414-416: Add a figure(s) for this! It's an interesting result but I don't see it visually

- Line 433: “estuarine biogenic habitats”

- Lines 445-447: Consider looking into literature on edge effects and how fragmentation in seagrass beds influences predator-prey dynamics.

- Line 455: Again, since this is a major point of discussion there should be a clear visualization of these trends.

- Lines 529-531: From what I understand based on figure 3, juvenile spotted seatrout were found more frequently in areas with a high proportion of saltmarsh during dry conditions. Isn't this the opposite of what's being stated here?

- Line 536: Are you only referring to biogenic habitat use here? What about distance from the inlet (fig 3 C,F)?

- Lines 540-541: I'm not seeing this result in figure 3. I see high probability of presence in seagrass habitat during wet AND dry conditions for the smallest size class (panel A), and lower presence in seagrass during dry conditions for the next size class (panel D).

- Lines 547-550: Develop this idea a bit more - they seemed to move down-estuary (closer to inlet) under wet conditions, and there appears to be less saltmarsh present near the inlets. Make that link to the spatial arrangement of habitat. Also, why would they move down-estuary under high inflow? Is it physiological (salinity stress)? Physical transport?

Figures:

- Fig 1 could add stream gauge location

- I like the predictive maps!

- Figures 3 and 4 are well done, but I think we’re missing some important results like the habitat richness. Consider adding to these figures or creating additional figures to clearly show the habitat richness effects.

- In figures 3 and 4, what freshwater inflow variable are we looking at? 1,3,6,12 month? The methods section is not super clear in describing how the figures were created (see comments in that section)

Reviewer #2: This paper presents a detailed analysis of a long-term dataset to describe drivers of variation in juvenile seatrout distribution in Apalachicola Bay, Florida. The paper is well written, clear, and concise. Although focused on one species in one large estuary, I think the paper is of wide interest to coastal ecologists, and provides a framework for this type of long-term data analysis. I think it is a solid paper and I have just a few comments for the authors to consider.

General comments

No mention of direct effects of salinity on juvenile trout distribution. At some level of FW inflow, salinities may become too low for small juveniles. I think this should be at least mentioned. E.g. the results at L333-336 suggest during high flows, small juveniles were not found up the bay far from the inlets, presumably in areas where salinity would be depressed. Perhaps this simply reflects salinity being too low for them at those times? Similarly the results at L337-339.

Context used here is not quite the same as defined by Bradley et al. 2020. Here the authors refer to “seascape context” when discussing the number and extent of other habitats in the surrounding seascape. This is a component of context described by Bradley et al. but context also includes other factors such as tide range, climate, general rainfall regime, geomorphology. Of course, in the current study, many of these other factors are constant, since the whole study is in one large estuary. In the current study, all factors considered in aim 2 (L 125-127) would be part of the “context” of each sampling location. It’s a pretty minor point.

Minor comments:

L56: should “represent” be “support”?

L181: how stable are the seagrass beds in the bay? How much might they have changed since 2010, or over the sampling period? I think 2010 is a good time point to use, since it is in the middle of the time period of nekton sampling, but some mention of the stability of seagrass in the bay would be helpful.

L197: can you describe what least-cost distance is? Is it the shortest distance via water from a point to the nearest inlet?

The modeling approach is quite complex, but the authors have done a good job describing it, and it seems appropriate for the data.

Reviewer #3: This paper analyzes the relationship between estuarine habitat, freshwater inflow, and juvenile spotted seatrout occurrence in Apalachicola Bay, FL. The study uses a seascape approach to quantify habitat surrounding sampling sties from a long term fisheries independent monitoring program and river flow at various time lags in a GLM framework. The main findings were that occurrence of small juveniles is positively correlated with seagrass area and river flow 3 months prior; and occurrence of large juvenile is related to availability of nearby habitats. The paper is well written, and the methods are clear and technically sound. I recommend it for publication after addressing my concerns and comments below.

Introduction

Line 80 Provide reference for recreational importance statement, and some values for amount of effort or landings from recreational fishing survey (MRIP)

Line 112 A recent study by Nehemiah et al (2022, 10.1002/mcf2.10201) found that temperature and adult abundance during the spawning season were the best predictors of year class strength, but they did not find a relationship with river flow. The contrasting results of various studies highlights the challenges with understanding this relationship.

Methods

Line 170 When is the spawning season? That should be stated somewhere, for context.

Line 174 Provide some information on samples sizes from this dataset.

Line 183 Habitat has changed drastically over this time period. Oysters in particular have declined rapidly over the last few years. It’s unlikely the habitat maps would ground truth very well with the FIM samples at fine spatial scales. Did the authors attempt to compare the reported habitat types at each FIM sample location with the maps? This is probably the biggest issue with the analysis, yet the assumption made here of static habitat layers is not addressed, in discussion and/or sensitivity analyses.

Line 214 Is the proportion of months in each flow stage treated as a continuous variable in the models? It seems to be an odd way to characterize flow, and the variable can take a limited number of values based on number of months.

Line 230 How did you decide which of the correlated variables to omit?

Line 233 So if it’s binary wet or dry, how are normal conditions represented, a zero for both?

Results

Line 330 What is meant by prolonged?

Line 332 Is it possible that seatrout are moving to optimal salinity conditions. Can you distinguish between preference for marsh habitat or preference for certain salinity ranges during dry years?

Discussion

Line 410 I don’t think the first two paragraphs are necessary, as they seem to reiterate the results.

Line 443 Perhaps this paper is relevant here: Barry SC, Hyman AC, Jacoby CA, Reynolds LK, Kowalewski M, Frazer TK. Variation in seagrass-associated macroinvertebrate communities along the Gulf Coast of Peninsular Florida: an exploration of patterns and ecological consequences. Frontiers in Marine Science. 2021 Mar 4;8:596966.

Line 447 The word vulnerable is not the best choice. Maybe ‘responsive’ is better.

Line 493 See Sinnickson et al. (https://doi.org/10.1007/s10021-023-00845-1) for some thoughts on time lags between flow and biomass at different trophic levels.

Line 522 Sinnickson et al (2023) showed a slightly longer lag from phytoplankton to benthic inverts than phyto to zoo under simulated top-down forcing.

Line 529 This statement seems contradictive to the finding that seatrout occurrence is higher in marsh habitat during dry conditions.

Table 1 How are normal flows represented in the models? That wasn’t clear in the methods. Is it implicit, as 1-(Wet+Dry)?

Fig 4 Missing letter A in caption.

6. PLOS authors have the option to publish the peer review history of their article (what does this mean?). If published, this will include your full peer review and any attached files.

Reviewer #1: **Yes: **Mariah Livernois

Reviewer #2: No

Reviewer #3: No

---

## [Author Response · Author response to Decision Letter 0]

6 Oct 2023

Dear Dr. Fujiwara:

In response to reviewers’ helpful comments, we have revised our manuscript titled: “The influence of freshwater inflow and seascape context on occurrence of juvenile Spotted seatrout Cynoscion nebulosus across a temperate estuary” submitted to PLOS One. The reviewers recommended that we give a more thorough description of how the inflow variables were calculated and included in the models. In response, we have added a more detailed description of inflow variables in the methods section (Lines 205 - 219). All line numbers in this letter refer to the revised manuscript without tracked changes.

Reviewer #1 also pointed out that a figure was needed to illustrate relationships between occurrence patterns and habitat richness. We have created new panels in Fig 3 and 6 to show this relationship.

We addressed the need to include more information about the stability of habitat coverages used in our study on Lines 170 – 186 in the methods section.

We have addressed all of the comments in this letter. Reviewer comments and our detailed responses are provided below.

All data used in this analysis can be accessed here: https://f50006a.eos-intl.net/F50006A/OPAC/Details/Record.aspx?BibCode=5847989

Reviewer #1:

There is some confusion about how the freshwater inflow variables were determined and incorporated into the models.

We have reworded and added more details to description in the methods section of how the time lags were determined (Lines 226 – 240) and used in the models (Lines 325 - 327). 

The time lag component was a bit unclear, and didn’t come across well in the figures

In addition to the more detailed description in the methods as mentioned above, we also included which time lag for the inflow variables were used in the captions of the line plots (Figs 3 and 6).

The authors also make some statements and conclusions regarding the effect of freshwater inflow on presence probability, but from what I understand the results only show the effect of freshwater inflow on spatial distribution of seatrout and their use of biogenic habitats. This wording needs to be clarified throughout the manuscript and possibly visually clarified via figures. 

We have ensured that any discussion of how freshwater inflow affected seatrout occurrence highlights the context-dependent nature of those effects.

The figures could benefit from some reworking, since I found that they didn’t clearly visualize some of the patterns the authors discuss (especially habitat richness). I also wonder if more detailed figures would help alleviate confusion regarding the freshwater inflow variables.

We have added panels to Figures 3 and 6 to help readers visualize the influence of habitat richness on seatrout occurrence.

Line 23: suggest “recreational sportfish”

Added "sport"

Line 26: “We used”

corrected

Lines 34-35: I'm not sure how we can determine a positive relationship between presence and inflow based on the figures provided. From what I can gather based on the figures, the models assess the effect of inflow on habitat use patterns, not the effect of inflow on presence probability. Reconsider this wording.

We agree and have omitted the word "positively" from this sentence and instead simply state: "The probability of encountering the two smallest juvenile seatrout size classes (15 – 50mm and 51 – 100mm SL) was also related to freshwater inflow conditions, particularly within a 3-month period prior to and including peak recruitment."

Line 43: suggest changing “larger sized fishes” to “larger juveniles”

We corrected this (Line ##).

Lines 57-63: Could be informative to add some text here linking freshwater inflow patterns with these habitats. Habitat-forming plants and animals also respond to changes in freshwater inflow, meaning the two concepts (freshwater flow/salinity and availability of structured habitat) are inherently linked. Furthermore, the results suggest use of different habitats depends on inflow conditions, so the two ideas should be more cohesive instead of separate (as written).

We have added text to explain that riverine flows can affect the spatial distribution of structured habitats (Lines 57 – 61). 

Lines 63-65: With the addition of text above, this could become a new paragraph about habitat richness.

In response, we have added more details about habitat richness to the paragraph introducing the importance of nearby habitat availability, both richness and area (Lines 71-72).

Line 79: “popular recreational fishery species”

We changed the wording here as requested.

Line 80: “live out life” is odd wording, suggest something like “spend their lifespan” or “are resident within”

We changed the wording here as requested to “Spotted seatrout complete their life cycle within their natal estuary” Line 78

Line 85: I suggest staying consistent as possible with “juvenile spotted seatrout” throughout the manuscript. (Here it says juveniles of seatrout, but later says juvenile spotted seatrout).

We have searched through the manuscript and modified all occurrences of "juveniles of seatrout" to "juvenile seatrout" to be consistent.

Line 120-121: Maybe give context here for peak recruitment – which months/time of year?

We added details on the months of peak recruitment for both smaller and larger juveniles. (Lines 181 – 185)

Lines 177-184: Has there been any change in saltmarsh/seagrass/oyster coverage across the study period (2001-2018), and do you think the coverage you have is reasonably close to the coverage across that timescale? I would suggest adding a justification for the mismatch between habitat and fish data. I appreciate the challenges with getting high res habitat data, so I assume you did the best you could, but a quick discussion of potential caveats would be good to include.

We used the best available habitat maps for the study area and have added references that analyzed the habitat maps over time and found that seagrass and salt marsh coverages were stable over the time scale of this study (Lines 194 – 196). We also provided more details about the oyster maps and explain that the habitat data includes both living and dead oysters, and dead oysters likely continue to provide protective habitat for small fishes (Lines 199 - 202).

Line 187: Add a little more detail about QA/QC – not sure what standard means

We provided more details regarding the quality assessment on Lines 205 – 207: “We performed quality control checks on all GIS layers including map projection definition, verifying attributes were within described ranges, and topological checks to verify there were no overlapping features.”

Lines 194-198: Could do this same calculation for distance from the major freshwater inflow source. Looking at Fig 1 there are collection points way out East in the bay that are far from the river, and others that are immediately adjacent to the river. I would expect that those locations would not experience the same magnitude of influence of inflow, right? Giving all locations the same inflow value seems like you might be missing some important spatial patterns of inflow effect.

We agree this is an interesting idea and did calculate the distance to nearest river and have added details describing this in the methods (Lines 220 – 221). However, this variable was highly correlated with other variables that we included in the model, particularly distance to nearest inlet, which we ultimately selected for the analysis. Hence, spatial patterns were accounted for to the greatest extent possible using distance to inlet and latitude and longitude variables. 

Lines 206-208: This sentence is confusing - please clarify wording and provide a more specific citation

We added a citation to the National Water Information System in Line 231. In addition, we have reworded, clarified, and added details to our explanation of the inflow variable calculations (Lines 234 – 243).

Lines 209-211: Would be helpful to add a marker for the stream gauge location on Fig 2 to give spatial context

We have added a point on Figure 2 to show the location of the stream gauge.

Lines 211-212: This is essentially a time lag, right? This determines whether recent or past conditions affect the fish more or less? Please clarify the wording of this.

We have reworded, clarified, and added details to our explanation of the time lag inflow variable calculations (Lines 234 – 243). It now reads: “We characterized freshwater inflow conditions over several time lags (1-month, 3-months, 6-months, 12-months), prior to and including the month of fish sampling, by calculating the proportion of months within each time lag that were classified as below normal streamflow (% low flow or dry conditions), normal (% normal flow) and above normal streamflow (% high flow or wet conditions).” Lines 211-216

Lines 213-215: How is this incorporated into the analyses? I'm having a hard time understanding what the freshwater inflow predictor variables actually look like. I would think that for each seine data point, there would be a classification of wet, dry, or normal during the month prior, and 3, 6, and 12 months prior - is that correct? I think there needs to be a clearer explanation of how these variables truly operate within the models.

We have clarified that "For each time lag, normal flow conditions served as the statistical baseline (i.e., % normal = 1 – % wet + % dry); hence, subsequent logistic regression models (see Statistical analysis, below) only included flow variables associated with wet and dry conditions as predictor variables. " (Lines 216 – 219)

Lines 272-273: This wording is confusing - the sentences above make it sound like AUC and Brier scores were calculated for each model (without testing vs. training datasets). Consider changing the order of how this process is presented, starting with the splitting of data into testing and training, followed by AUC and Brier score calculations, and ending with how you summarized the performance results.

We have omitted the sentence stating "We note that we used all available data for each size class to fit the models described in the preceding paragraph." and revised the next sentence (L275-279) to read: To assess the performance of each model, we split the data for each size class into separate training (80% of data) and testing data (20% of data) sets, re-fit the models in each size class’s confidence set to the training data, and calculated testing and training AUC and Brier score statistics for both the training and testing data."

Lines 282-283: Does this refer to calculated predictions of encounter probabilities across the range of values for each predictor variable? The wording in this section is hard to follow

We have clarified this section (Lines 285 - 311) which now reads: 

"Model predictions

Prediction maps. - We calculated estuary-wide, model-averaged predictions of encounter probabilities for each of the four size classes using AICc weights (re-scaled to sum to one across models in each confidence set) to weight the contribution of each model’s predictions to the model-averaged predicted mean encounter probabilities. For these predictions, we used existing estuary-wide spatial layers (raster data used to create model covariates) for the seascape- and estuary-scale predictors. We exported the model predictions from R as GeoTiff files into a GIS (ArcGIS, ESRI). In GIS, we clipped the spatial extent of prediction grids to near-shore shallow waters (depths ≤ ~1.8 m) that best-represented the extent of fish sampling locations. To demonstrate the influence of varying freshwater inflows on the predicted encounter probabilities, we calculated the model-averaged predictions for each size class under wet (the wettest observed freshwater inflows), normal, and dry (the driest observed freshwater inflows) hydrologic conditions. 

Prediction line plots. - In addition to the estuary-wide predictions, we used line plots to demonstrate the influence of changing freshwater inflows and seascape- and estuary-scale variables by calculating mean encounter probabilities and 95% confidence intervals, again under three levels of freshwater inflow (wettest, normal, and driest observed conditions). Predictions for the line plots were calculated separately over the observed range of seagrass coverage (all size classes), over the observed range of salt marsh coverage (only ≤50mm and 51-100mm size classes), over the observed range of distance to the nearest major inlet (all size classes), and over the observed range of habitat richness (all size classes). For simplicity, for the line plot predictions we used the top confidence set model in which each predictor variable occurred to make predictions (i.e., these were not model averaged predictions)."

Lines 285-287: Don't the models already include values for the habitat variables around each sample location? Wouldn't the range of those values be used for prediction? This makes it sound like it’s predicting spatially, but I'm not sure why that would be necessary here (since it occurs in the prediction maps section)

Please see the above response that addresses this issue. This was revised on Lines 285 – 311 in the manuscript.

Lines 287-290: I assume this is what we're seeing in Figs 3 and 6, but it isn't super clear. Predicted encounter probabilities were calculated across the possible range of each predictor variable for each inflow condition, correct?

Please see the above responses (Line 282 – 283 and Line 285-287) that address this issue. We created a paragraph titled “Prediction lines” and the description of this was revised on Lines 298 – 307 in the manuscript.

Line 293: How does the above paragraph describing model prediction link to these maps. I see here that you've exported a spatial map of predictions, but then it's later stated (last sentence of this paragraph) that you only used the top model for this. Aren't the predictions coming from the weighted/averaged models above? Please clarify this section and the previous paragraph to clearly state what models and predictions relate to each figure (effects vs. maps)

Please see the above responses (Lines 282 – 283, 285-287, and 287-290) that address this issue. We created a paragraph titled “Prediction maps” and the description of this was revised on Lines 286 – 297 in the manuscript.

Line 329: This isn't something I can easily distinguish from the figures - There's no figure that shows the relationship between habitat richness and spotted seatrout presence

We have modified Figures 3 and 6 to demonstrate the influence of habitat richness.

Line 330: Same here, no visualization of the oyster reef trend

Unfortunately, there were too many variables to create plots for each one. Although we did not add an oyster plot, we did describe the relationship between oyster coverage and seatrout occurrence in the results section (Lines 334 and 383) and in Tables S1 – S4 in the supporting information.

Lines 330-332: I suggest adding figure labels (including panel labels) here to distinguish exactly what fig the reader should look at to see these patterns.

After carefully considering this comment, we felt that the figure panels were clear enough without additional labels.

Line 334: What does prolonged mean in this context? This word is used throughout the results but in relation to all time lags (1,3,6,12 months), which doesn't make sense to me. Define what "prolonged" means in the context of the freshwater flow variables.

Here, we mean inflow conditions were predominantly wet, normal, or dry over the window interval. We have removed the word “prolonged” throughout the manuscript and replaced it with “predominantly” as it better describes the inflow conditions. (see Line 385 and throughout the manuscript)

Lines 414-416: Add a figure(s) for this! It's an interesting result but I don't see it visually

We have modified Figures 3 and 6 by creating new panels for habitat richness for the two smaller size classes in Fig 3 (D and H), and also for the larger sized seatrout in Fig 6 (C and F).

Line 433: “estuarine biogenic habitats”

Have changed the wording to reflect this suggestion (Line 437 and elsewhere throughout the manuscript).

Lines 445-447: Consider looking into literature on edge effects and how fragmentation in seagrass beds influences predator-prey dynamics.

We have referenced a review paper of the influence of edge effects in predation in seagrass habitat (Line 445).

Line 455: Again, since this is a major point of discussion there should be a clear visualization of these trends.

We have modified Figures 3 and 6 by creating new panels for habitat richness for the two smaller size classes in Fig 3 (D and H), and also for the larger sized seatrout in Fig 6 (C and F).

Lines 529-531: From what I understand based on figure 3, juvenile spotted seatrout were found more frequently in areas with a high proportion of saltmarsh during dry conditions. Isn't this the opposite of what's being stated here?

We explained this counterintuitive result in the discussion (Lines 551 - 560) as a limitation of the seine gear used in the study. Fish inhabiting the inundated marsh during wet conditions are unavailable to be captured by the seine as it can only sample along the marsh edge. During dry conditions, the marsh is frequently dry and the fishes are pushed into unvegetated areas where they are able to be captured by the seine.

Line 536: Are you only referring to biogenic habitat use here? What about distance from the inlet (fig 3 C,F)?

We discuss biogenic habitat use as well as spatial distribution relative to the inlet. We added "biogenic" to clarify in Line 537. We also have mentioned the weak evidence found for a spatial shift in juvenile encounter probability toward the inlet during wet conditions when juveniles may moved downstream from marsh areas (Lines 547-551).

Lines 540-541: I'm not seeing this result in figure 3. I see high probability of presence in seagrass habitat during wet AND dry conditions for the smallest size class (panel A), and lower presence in seagrass during dry conditions for the next size class (panel D).

In response to this comment, we double checked results and corrected the text in Lines 544-545 to state that the small juveniles were encountered more frequently in wet conditions, we had erroneously stated "dry" conditions in the previous draft. Although Figure 3 shows a similar relationship between seagrass and presence of seatrout in wet and dry conditions, this relationship was not significant for dry conditions.

Lines 547-550: Develop this idea a bit more - they seemed to move down-estuary (closer to inlet) under wet conditions, and there appears to be less saltmarsh present near the inlets. Make that link to the spatial arrangement of habitat. Also, why would they move down-estuary under high inflow? Is it physiological (salinity stress)? Physical transport?

We developed the idea that the heavy inflow during wet conditions resulted in a spatial shift downstream for the smallest juveniles and moved them away from salt marshes that mainly occurred near the rivers. We also mention that the spatial shift toward the inlet may be caused by transport (flushing) or osmoregulatory stress. 

Fig 1 could add stream gauge location

The stream gauge location was added to figure 2.

I like the predictive maps!

Thank you!

Figures 3 and 4 are well done, but I think we’re missing some important results like the habitat richness. Consider adding to these figures or creating additional figures to clearly show the habitat richness effects.

Glad to hear that Figures 3 and 4 resonated. We have modified Figures 3 and 6 to demonstrate the influence of habitat richness.

In figures 3 and 4, what freshwater inflow variable are we looking at? 1,3,6,12 month? The methods section is not super clear in describing how the figures were created (see comments in that section)

The inflow variable being presented in figures 3 and 6 depends on the size class. We stated in the Methods (more clearly now, hopefully, Lines 298 – 307) that these figures are based on predictions from the best-approximating model for each size class. For the predictive maps, however, we used model averaged predictions (Lines 286 - 297). We also added which flow variable was used for prediction to each figure legend. This information is also available in the supporting information Tables S1 – S4).

Reviewer #2 Comments:

No mention of direct effects of salinity on juvenile trout distribution. At some level of FW inflow, salinities may become too low for small juveniles. I think this should be at least mentioned. E.g. the results at L333-336 suggest during high flows, small juveniles were not found up the bay far from the inlets, presumably in areas where salinity would be depressed. Perhaps this simply reflects salinity being too low for them at those times? Similarly the results at L337-339.

We have expanded our discussion of potential effects of salinity and salinity fluctuations on the distribution of small juveniles (Lines 498 – 501) by discussing the potential role of rapidly changing salinity conditions that may cause osmoregulatory stress on larvae and juveniles. We discuss the weak evidence that we found that during wet conditions, the two smaller size classes were less likely to occur in areas located farther from the nearest inlet to the Gulf of Mexico. We mention that these areas may have also undergone a reduction in salinity with increased inflow during wet conditions that could have caused osmoregulatory stress to the early life stages of seatrout (Lines 506 – 511).

Context used here is not quite the same as defined by Bradley et al. 2020. Here the authors refer to “seascape context” when discussing the number and extent of other habitats in the surrounding seascape. This is a component of context described by Bradley et al. but context also includes other factors such as tide range, climate, general rainfall regime, geomorphology. Of course, in the current study, many of these other factors are constant, since the whole study is in one large estuary. In the current study, all factors considered in aim 2 (L 125-127) would be part of the “context” of each sampling location. It’s a pretty minor point.

We agree this is a minor point but appreciate the reviewer’s attention to such detail. Bradley et al. (2020) define seascape context as "unique properties of a location of interest (at any scale) that determine ecological functionality." We chose a scale of 400-m to examine seascape context in relation to nearby habitat types. The term "seascape context" is derived from "landscape context" that has a long history of use in the field of landscape ecology. The metrics used to measure landscape context of nearby habitat types in this study are used extensively in the field of landscape ecology. As such, although we understand that the spatial scale of factors defining “seascape context” can vary depending on study objectives, we have been clear about how seascape context was defined in this study and are confident that our use of the term is similar what appears in the broader literature.

L56: should “represent” be “support”?

We replaced the word as suggested (Line 56).

L181: how stable are the seagrass beds in the bay? How much might they have changed since 2010, or over the sampling period? I think 2010 is a good time point to use, since it is in the middle of the time period of nekton sampling, but some mention of the stability of seagrass in the bay would be helpful.

In response to this comment, we have added some details and references in the methods to describe the stability of seagrass coverage over the sampling period. We used the best available habitat maps for the study area. I have added some references that analyzed the habitat maps over time and found that seagrass and salt marsh coverages were stable over the time scale of this study (Lines 194 – 196). 

L197: can you describe what least-cost distance is? Is it the shortest distance via water from a point to the nearest inlet?

We have reworded the description of the least-cost distance to better explain that it is the shortest distance from each grid cell in the study area to the nearest inlet using the shoreline as a barrier so that distances are calculated over the water only. (Lines 193 – 197)

The modeling approach is quite complex, but the authors have done a good job describing it, and it seems appropriate for the data.

Thank you!

Reviewer #3 Comments:

Line 80 Provide reference for recreational importance statement, and some values for amount of effort or landings from recreational fishing survey (MRIP)

We added a reference to a NMFS report to support this statement (Line 77-78).

18. Pritchard ES. Fisheries of the United States 2004. National Marine Fisheries Service; 2006 Jan 10.

Line 112 A recent study by Nehemiah et al (2022, 10.1002/mcf2.10201) found that temperature and adult abundance during the spawning season were the best predictors of year class strength, but they did not find a relationship with river flow. The contrasting results of various studies highlights the challenges with understanding this relationship.

We have added a reference to this study in the introduction (Lines 98 and 104) to explain that relationships between inflow and juvenile seatrout distribution appear to vary depending upon temporal scale at which inflows are measured in various studies including Nehemiah et al. 2022. 

Line 170 When is the spawning season? That should be stated somewhere, for context.

In the methods, we included the peak spawning season in Apalachicola Bay which occurs from April to August according to Brown-Peterson et al. 2002 (Line 161). 

Line 174 Provide some information on samples sizes from this dataset.

In the methods, we included sample sizes (Lines 162 – 165). 3,094 unique samples for the two smaller juvenile size classes (15 - 50mm SL and 51 – 100mm SL) and 1,782 unique samples for the larger two juvenile size classes (101 – 150mm SL, 151 – 200mm SL). The full datasets are provided here: https://f50006a.eosintl.net/F50006A/OPAC/Details/Record.aspx?BibCode=5847989

Line 183 Habitat has changed drastically over this time period. Oysters in particular have declined rapidly over the last few years. It’s unlikely the habitat maps would ground truth very well with the FIM samples at fine spatial scales. Did the authors attempt to compare the reported habitat types at each FIM sample location with the maps? This is probably the biggest issue with the analysis, yet the assumption made here of static habitat layers is not addressed, in discussion and/or sensitivity analyses.

We have added some more information about the oyster maps used in the analysis (Lines 175 - 182). Although it is true that coverage of living oyster beds in Apalachicola Bay has declined dramatically in recent years, oyster shells have remained and likely continue to provide habitat for small fishes by providing protection from predation. Therefore, oyster maps used in this study are based on coverage of oyster habitat (both living and dead). In addition, spatial coverages of habitat are collected on a broader spatial scale than fish sampling point locations. Therefore, it would be difficult to estimate in the field the area of oyster beds around a 400-m radius around each sample site to compare habitat area calculated from maps with field observations of oyster habitat.

Line 214 Is the proportion of months in each flow stage treated as a continuous variable in the models? It seems to be an odd way to characterize flow, and the variable can take a limited number of values based on number of months.

Yes, it's the proportion of months within the time lag that the conditions were observed. We could have used the count of the number of months as a continuous predictor as well but instead chose to express this variable as a proportion. This is simply a choice we made regarding how to express the influence of flow conditions, but we view this as an arbitrary scaling choice that would not affect overall estimates and, hence, inferences made from the models. We have clarified this in the methods. (Lines 205 – 219)

Line 230 How did you decide which of the correlated variables to omit?

We state on Lines 233- 235 that to avoid multicollinearity we did not include any predictor variables with a Pearson correlation coefficient greater than |0.5|, which is our experience is more than adequate for avoiding collinearity problems. When two variables were found to be colinear, we selected the variable that was most in line with our modelling objectives.

Line 233 So if it’s binary wet or dry, how are normal conditions represented, a zero for both?

Yes, the one-month window was essentially a dummy variable with 3 possible outcomes, so for a given flow variable (e.g., 3-month window) normal flows represented the reference condition and could be calculated as 1 - Wet + Dry. We have clarified this in the methods (Lines 216 -219).

Line 410 I don’t think the first two paragraphs are necessary, as they seem to reiterate the results.

These paragraphs (Lines 414 – 432) are a general summary of the results, which we believe is helpful for readers to understand the main discussion points resulting from the study.

Line 443 Perhaps this paper is relevant here: Barry SC, Hyman AC, Jacoby CA, Reynolds LK, Kowalewski M, Frazer TK. Variation in seagrass-associated macroinvertebrate communities along the Gulf Coast of Peninsular Florida: an exploration of patterns and ecological consequences. Frontiers in Marine Science. 2021 Mar 4;8:596966.

Thank you for bringing this paper to our attention as it covers concepts regarding nutrients and benthic resources in seagrasses in a nearby estuarine system. On Lines 444-445, we were making the point that there is generally a higher abundance of benthic prey in seagrass versus unvegetated areas. Due to the wealth of literature showing this trend, we went with a review of the literature by Heck et al. 2008 to reduce the need for additional citations.

Line 447 The word vulnerable is not the best choice. Maybe ‘responsive’ is better.

We agree and have made this change (Line 476 ).

Line 522 Sinnickson et al (2023) showed a slightly longer lag from phytoplankton to benthic inverts than phyto to zoo under simulated top-down forcing.

Thank you for bringing the Sinnickson et al. (2023) paper to our attention. We included it as a reference to support our statement (Line 521-522) regarding the likely longer lag time between discharge fluctuations and response by benthic prey populations versus a shorter time lag between discharge and planktonic prey concentrations.

Line 529 This statement seems contradictive to the finding that seatrout occurrence is higher in marsh habitat during dry conditions.

We explained this counterintuitive result in the discussion (Lines 551 - 560) as a limitation of the seine gear used in the study. Fish inhabiting the inundated marsh during wet conditions are unavailable to be captured by the seine as it can only sample along the marsh edge. During dry conditions, the marsh is frequently dry and the fishes are pushed into unvegetated areas where they are able to be captured by the seine.

Table 1 How are normal flows represented in the models? That wasn’t clear in the methods. Is it implicit, as 1-(Wet+Dry)?

Yes, the one-month window was a binary variable with 3 possible outcomes, so for a given flow variable (e.g., 3-month window) normal flows represented the reference condition and could be calculated as 1 - Wet + Dry. The 3, 6, and 12-month windows follow the same logic except those values were not binary and could take any value between 0 and 1. We have clarified this in the methods (Lines 211 – 219).

Fig 4 Missing letter A in caption.

We have added the missing A in the caption for Fig 4.

---

## [Editor Report · Decision Letter 1]

27 Oct 2023

The influence of freshwater inflow and seascape context on occurrence of juvenile Spotted seatrout Cynoscion nebulosus across a temperate estuary

PONE-D-23-22016R1

Dear Dr. Whaley,

We’re pleased to inform you that your manuscript has been judged scientifically suitable for publication and will be formally accepted for publication once it meets all outstanding technical requirements.

Kind regards,

Masami Fujiwara, PhD

Academic Editor

PLOS ONE
---

## [Editor Report · Acceptance letter]

15 Nov 2023

PONE-D-23-22016R1 

The influence of freshwater inflow and seascape context on occurrence of juvenile Spotted seatrout *Cynoscion nebulosus* across a temperate estuary 

Dear Dr. Whaley:

I'm pleased to inform you that your manuscript has been deemed suitable for publication in PLOS ONE. Congratulations! Your manuscript is now with our production department. 

Kind regards, 

on behalf of

Dr. Masami Fujiwara 

Academic Editor

PLOS ONE